**Resource**

# Proteolytic profiling of human plasma reveals an immunoactive complement C3 fragment

Fatih Demir [ID][1✉], Elina Kovalenko [ID][1], Moritz Lassé[2,3], Esben B Svenningsen[4], Jens M Bernth Jensen[5,6], Anja M Billing [ID][1], Kathrin Groeneveld[7], Arvid Hutzfeldt [ID][2,3], Lars Nilges[2,3], João P L Guerra [ID][8], Krzysztof J Pietrzak-Lichwa [ID][8], Yifan Tan[1], Elizabeth Colby[9], Annette G Hansen[1], Naziia Kurmasheva[1], David Olagnier [ID][1], Dongwoo Choi[1], Mika M Richter[2,3], Sandra D Laufer [ID][2,3], Fabian Braun[2,3,10], Sally A Johnson[11], Marcus Krüger [ID][12], Tobias B Huber [ID][2,3], Elion Hoxha[2,3], Oliver M Steinmetz[2,3], Ralf Mrowka[7,13], Simon Melderis [ID][2,3], Moin A Saleem[9], Thomas B Poulsen [ID][4], Gregers R Andersen [ID][8], Steffen Thiel [ID][1], Anne Troldborg[1,14,15] & Markus M Rinschen [ID][1,2,3,11,15✉]

## Abstract

Dysregulated proteolysis is central to autoimmune pathogenesis. The complement cascade, a major protease network, generates fragments that modulate immunity and tissue injury. We developed a scalable blood plasma N-terminomics workflow that markedly expands detection of proteolytic events in vitro and in vivo. Applied to 143 systemic lupus erythematosus (SLE) patients, Multi-Omics Factor Analysis (MOFA) linked N-terminal signatures to immunological and clinical heterogeneity. This revealed a previously unrecognized complement fragment, C3-LHF1, encompassing the C345C domain and rivaling, based on intensity detected by mass spectrometry, the abundance of canonical fragments like C3a and C3b. C3-LHF1 associated with renal function and remission in lupus nephritis, and exhibited dual functions: inhibiting classical and lectin complement pathways and acting as a partial IL6ST (gp130) agonist, independent of IL6Rα. In human kidney organoids, C3-LHF1 induced JAK/STAT3 signaling, amplified TNFα-driven CXCL10 secretion, and reduced podocyte marker expression, suggesting a role in tissue remodeling. These findings reveal unanticipated complexity in complement-mediated signaling and provide a comprehensive atlas of protein N-termini in human plasma, which enables discovery of novel immunoregulatory mechanisms and therapeutic targets in inflammatory disease.

**Keywords** SLE; SHUNTER; N-terminomics; Proteolysis; LHF
**Subject Categories** Immunology; Post-translational Modifications & Proteolysis

## Introduction

Proteolysis, the generation of protein fragments, is a fundamental line of immune defense, driving processes such as pathogen clearance (Meyer et al, 2021), antigen processing (Harada et al, 2019; Colasanti et al, 2020), inflammatory propagation (Grillet et al, 2023), and host-microbiome homeostasis (Bülck et al, 2023). The human complement system is a prime example of a proteolytic cascade with high clinical relevance. The system comprises cell-surface receptors, regulators, and circulating complement proteins that combat microbial invaders, modulate immune responses, and remove senescent cells (Bekassy et al, 2022; Mastellos et al, 2024). Complement proteins circulate in inactive forms and are activated upon recognition of antibodies, microbial patterns, or another ligand. The expanding clinical relevance of the complement system is exemplified by an ever increasing number of available drugs that specifically target this system (Friščić et al, 2021; Zhong et al, 2023; Maffia et al, 2024; Revel et al, 2024). Due to the pharmacological accessibility of the human complement system, the spectrum of diseases thought to be at least partially complement-driven has expanded from rare diseases and autoimmune diseases (Weinstein et al, 2021) to a variety of conditions, including metabolic (Phieler et al, 2013; Tang and Yiu, 2020; Tan et al, 2020), cardiovascular-kidney (Campbell and Kahwash, 2020; Adamo et al, 2020; Kiss and

[1]Department of Biomedicine, Aarhus University, Aarhus, Denmark. [2]III. Department of Medicine, University Medical Center Hamburg-Eppendorf, Hamburg, Germany. [3]Hamburg Center for Kidney Health, Hamburg, Germany. [4]Department of Chemistry, Aarhus University, Aarhus, Denmark. [5]Department of Clinical Immunology, Aarhus University Hospital, Aarhus, Denmark. [6]Department of Molecular Medicine (MOMA), Aarhus University Hospital, Aarhus, Denmark. [7]Department of Internal Medicine III, Experimental Nephrology Group, University Hospital Jena, Jena, Germany. [8]Department of Molecular Biology and Genetics, Aarhus University, Aarhus, Denmark. [9]Bristol Renal, Bristol Medical School, University of Bristol, Bristol, UK. [10]Martin Zeitz Center for Rare Diseases, University Medical Center Hamburg-Eppendorf, Hamburg, Germany. [11]National Renal Complement Therapeutics Centre, Newcastle upon Tyne Hospitals National Health Service Foundation Trust, Newcastle upon Tyne, UK. [12]Cologne Excellence Cluster Cellular Stress Response in Aging-Associated Diseases (CECAD), and Faculty of Mathematics and Natural Sciences, Institute of Genetics, University of Cologne, Cologne, Germany. [13]Thuringian Innovation Center for Medical Technology Solutions ThIMEDOP, University Hospital Jena, Jena, Germany. [14]Department of Rheumatology, Aarhus University Hospital, Aarhus, Denmark. [15]These authors contributed equally: Anne Troldborg, Markus M Rinschen. ✉E-mail: fatih.demir@biomed.au.dk; m.rinschen@uke.de

Binder, 2022; Caravaca-Fontán et al, 2023), eye (Armento et al, 2021), neurodegenerative diseases (Schartz and Tenner, 2020; Tenner, 2020), and even cancer (Garred et al, 2021; West and Kemper, 2023), underscoring its broad translational relevance. For many diseases, complement modulators are in various phases of clinical and preclinical investigation (West et al, 2024).

Activation of the complement cascade occurs through three main pathways: i) the classical pathway, initiated by antigen-antibody complexes; ii) the lectin pathway, activated by carbohydrate patterns on microbial surfaces; and iii) the alternative pathway, activated continuously in plasma through spontaneous hydrolysis of C3 which but also essential for amplification of the cascade after initiation by the classical and lectin pathways. Upon activation, complement proteins C4, C3, and C5 undergo sequential proteolytic cleavage, generate potent effectors such as C3a and C5a, and initiate the formation of the membrane attack complex (MAC). C3a and C5a are anaphylatoxins that recruit immune cells to the infection site, thereby enhancing inflammation. The MAC, formed by the assembly of C5b through C9, perforates microbial membranes, which results in loss of energy production and direct osmotic lysis. Furthermore, the C3b opsonin and its degradation product iC3b enable phagocytosis of pathogens.

Complement activation is recognized as a hallmark of active Systemic Lupus Erythematosus (SLE), reflecting its pivotal role in driving inflammation and tissue injury in this disease (Weinstein et al, 2021; Manderson et al, 2004). Clinically, unrestricted complement consumption manifests as low serum C3 and C4 levels, a classic indicator of SLE activity, that is incorporated into disease activity indices like the SLEDAI (SLE disease activity index) score (Bombardier et al, 1992). This link is particularly pronounced in lupus nephritis (LN), the prototypic organ-threatening SLE manifestation (Sato et al, 2011; Bomback et al, 2016). The vast majority of patients with active renal disease display hypocomplementemia (Furie et al, 2020) and complement deposition within glomeruli is a defining feature of the immune complex–mediated kidney damage (Bajema et al, 2018). Beyond the consumption of

complement factors, elevated levels of split-products of complement proteins serve as dynamic biomarkers of disease activity. Fragments such as C3dg and C4d accumulate in active SLE, correlating closely with global disease activity (Martin et al, 2017; Troldborg et al, 2018a). Moreover, complement activation markers track clinical trajectories, for example, plasma C4d level correlates with glomerular C4d deposition and proteinuria in lupus nephritis, and it declines substantially in patients achieving remission while remaining elevated in non-responders (Martin et al, 2020). Collectively, these clinical and mechanistic insights establish that aberrant complement activation is tightly linked to SLE disease activity, especially LN, making SLE disease trajectory an obvious choice in which to study the proteolytic fragment generation of the complement cascade.

The high medical relevance of the complement system contrasts with the relatively static knowledge of its components generated by proteolysis (Fig. 1). Most textbooks describe a fixed number of well-characterized fragments, generated by a specific set of proteolytic events in a linear, stepwise cascade. Recently, through the development of novel mass spectrometry-based approaches, the view of linear proteolytic relationships has been challenged by the concept of a "proteolytic web" that is interconnected with multiple feedback loops (Klein et al, 2018; Kollet et al, 2024). The high throughput quantitation of N-termini has provided a unique window in the functional state of the proteome that is not amenable through affinity reagents. Established, mass-spectrometry-based N-terminomics workflows include COmbined FRActional DIagonal Chromatography (COFRADIC, Gevaert et al, 2003), Terminal Amine Isotopic Labeling of Substrates (TAILS, Kleifeld et al, 2010), High-efficiency Undecanal-based N Termini EnRichment (HUNTER, Weng et al, 2019) or subtiligase ligation (Mahrus et al, 2008). The aim of this study was to explore N-termini within human plasma in in vivo and in vitro settings and to identify circulating immunoactive protein fragments within the human autoimmune disease systematic lupus erythematosus (SLE). Using comprehensive analysis and clinically guided interpretation of the human N-terminome dataset (comprising

## Overview of the complement system

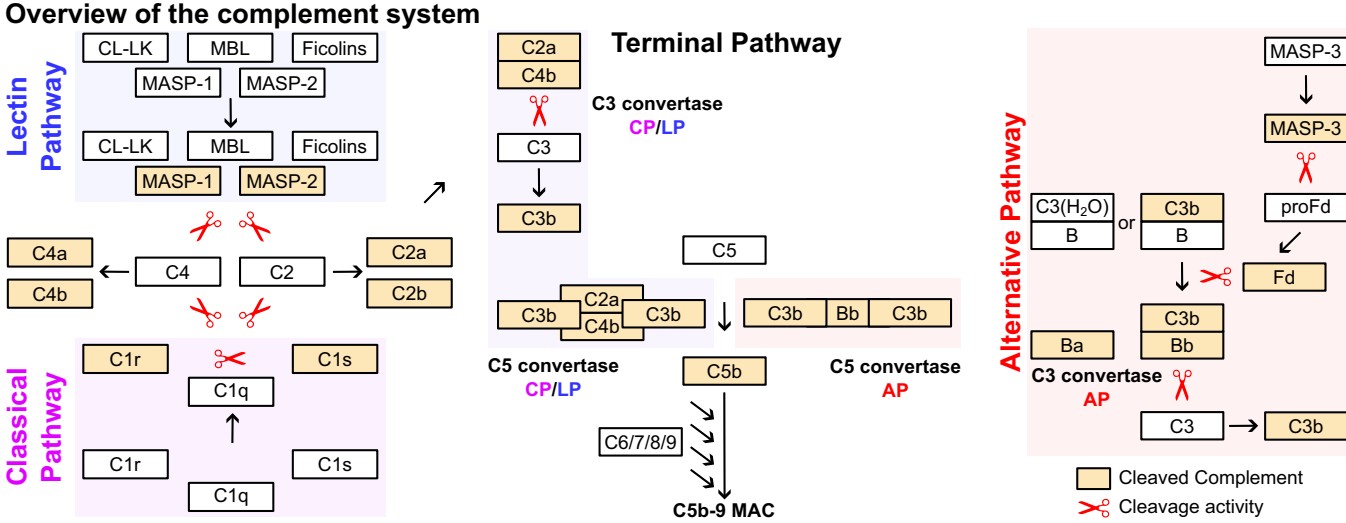

**Figure 1.** Schematic overview of the human complement system.

more than 11,000 N-termini on plasma proteins), we shed light on the proteolytic events governing critical biology and demonstrate unanticipated roles for novel fragments within the context of organ-specific disease.

# Results

## N-terminomics to detect protease-generated termini

In the HUNTER protocol (Weng et al, 2019) protease-generated new protein N-termini are retrieved as dimethylated N-terminal amines (Fig. 2A). By shortening the labeling time, we improved dimethylation efficiency (Appendix Fig. S1A, Dataset EV1), purity of the N-termini (Appendix Fig. S1B, Dataset EV2), and the number of totally identified N-termini (Appendix Fig. S1A–D, Dataset EV3) in human plasma. As suggested by previous results (Weng et al, 2019), data-independent acquisition enhanced (DIA) quantitation coverage of identifications (Appendix Fig. S1D, Dataset EV3). Unwanted side-reactions (lysine acetylation, guanidination, an unknown side chain reaction of 7.96 Da) with the HUNTER protocol were reduced (Appendix Fig. S1E,F, Dataset EV4). Thus, a substantially increased depth and purity of N-termini in human plasma was observed by streamlining the process and using more advanced data-independent acquisition-based mass spectrometry.

## Feasibility of N-terminomics to investigate complement inhibition

Next, our goal was to evaluate the plasma proteolytic signature of the widely used complement inhibitor eculizumab, which is a C5 binder. We analyzed patients with hemolytic uremic syndrome induced by Shiga toxin-producing *Escherichia coli* (STEC-HUS, Obrig and Karpman, 2012) and were part of the ECUSTEC trial (Ives et al, 2024). In this trial, complement inhibition by eculizumab was demonstrated, but clinical improvement was not consistent, making this a case to study eculizumab action on proteolysis. Plasma was obtained at baseline (day 1, defined as active disease) before, as well as days 8 and 30 after treatment with Eculizumab (300 mg on day 1 and day 8, Fig. EV1A). In total, 4789 endogenous protein N-termini and 514 N-termini from complement proteins were identified (Dataset EV5), with 476 N-termini from 130 gene products being significantly regulated ($|log_2FC| > 1$ and limma moderated $p$-value < 0.05) by eculizumab (Fig. EV1B,C). Globally, the abundance of N-termini was markedly different from the total protein level using bulk proteomics analysis of the same sample (Fig. EV1D,E, low Pearson's correlation coefficient, especially for complement proteins of $R = 0.15$ for days 8/1 and $R = 0.11$ for days 30/1). Eculizumab caused a reduction of C3, C9, and C4 N-termini (Fig. EV1F), as expected, suggesting that there was a reduction and feedback loop of C5 inhibition (Fig. EV1B,F). Specifically, eculizumab caused an early, significant increase in the N-terminus of C5 at I1381 ($log_2FC$ of +3.81 days 8/1, Fig. EV1B, marked in black). Structural mapping identified the I1381 cleavage site on C5 near a flexible loop of the MG8 and ANA domains released by C5 convertase, indicating that this proteolysis could cause partial unfolding of the MG8 domain (Fig. EV1G).

## Proteolytic landscape of plasma in a heterogenous lupus patient population

Recognizing our approach's potential to map the proteolytic landscape in human plasma, we aimed to leverage the heterogeneity of patients within an autoimmune disease cohort to uncover functional insights. We analyzed a cross-sectional, deeply phenotyped observational cohort of patients (Troldborg et al, 2018a, 2018b) with systemic lupus erythematosus (Fig. 2B), reflecting the heterogeneity of lupus phenotypes and complications ($n = 143$ patients, 91.6% female, mean age 45.3 ± 14.5 (SD), mean SLEDAI disease score 3.8 ± 2.1, 32.9% lupus nephritis, Dataset EV6). The clinical characteristics of the population were described before (Troldborg et al, 2018a, 2018b). We applied an automated workflow of our N-terminome enrichment using an Opentrons OT-2 lab robot. The number of proteolytic events per protein demonstrates that many cleavages occur in fibrinogen, albumin, and particularly in the C3 protein (Fig. 2C), the most abundantly cleaved protein of the complement system. When comparing N-termini in lupus patients to healthy controls ($n = 23$, mean age 34.3 ± 3.1), most SLE patients showed significant changes (Figs. 2D and EV3A; Dataset EV7). We also observed reduced proteolytic processing of inter-alpha-trypsin inhibitor heavy chain H1 (ITIH1) at A126 ($log_2FC$ SLE/CTRL −1.19), a protease inhibitor, and of PROS1 at F92 ($log_2FC$ SLE/CTRL −1.66), an anticoagulant protein with immunosuppressive abilities (Ubil et al, 2018). The bulk proteome was analyzed in parallel (Dataset EV7). N-termini abundance was in part independent of its respective protein on proteome level (Pearson's correlation coefficient $R = 0.038$, Fig. EV3B). Examining the extended cleavage specificity P6-P6' for the down- and up-regulated N-termini in SLE ($|log_2FC| > 0.35$, Fig. EV3C,D), we could observe a bias in arginine-directed cleavage, followed by threonine, serine and glutamine for the down-regulated N-termini (Fig. EV3C, $n = 879$ N-terminal cleavage windows). A more heterogeneous cleavage motif with arginine and alanine at the center was observed for the up-regulated N-termini (Fig. EV3D, $n = 754$ N-terminal cleavage windows) with a subtle leucine preference for the P6-P5 and P3-P2 positions. As complement C3 represented the hotspot for proteolytic processing events, we evaluated the identified C3 N-termini for global abundance in terms of $log_{10}$-transformed intensities for the CTRL and SLE samples (Fig. 2E) and could identify a very heterogeneous distribution for the N-termini spanning four orders of magnitude from $log_{10}$ intensities of 3.5–7.6 with a mean $log_{10}$ intensity of 5.10 in CTRL and 5.04 in SLE (Dataset EV7). The most abundant fragments/N-termini identified for C3 were C3(d)g (pos. 947/955), C3alpha (pos. 672/673), and C3b-c (pos. 569) which were already described (Troldborg et al, 2018a; Weinstein et al, 2021). Additionally, previously undescribed cleavages at pos. 1520 and 1514 (pre-proenzyme numbering) were very abundant (Fig. 2E). The cleavage at pos. 1514 was increased within the SLE patients ($log_2FC$ SLE/CTRL + 0.52 and a mean $log_{10}$ intensity of 6.53 in the SLE cohort).

## N-terminomic signals associated with clinical parameters and disease trajectory

Using Multi-omics-factor analyses (MOFA, Argelaguet et al, 2020), we analyzed the N-terminomic and proteomic dimensions of

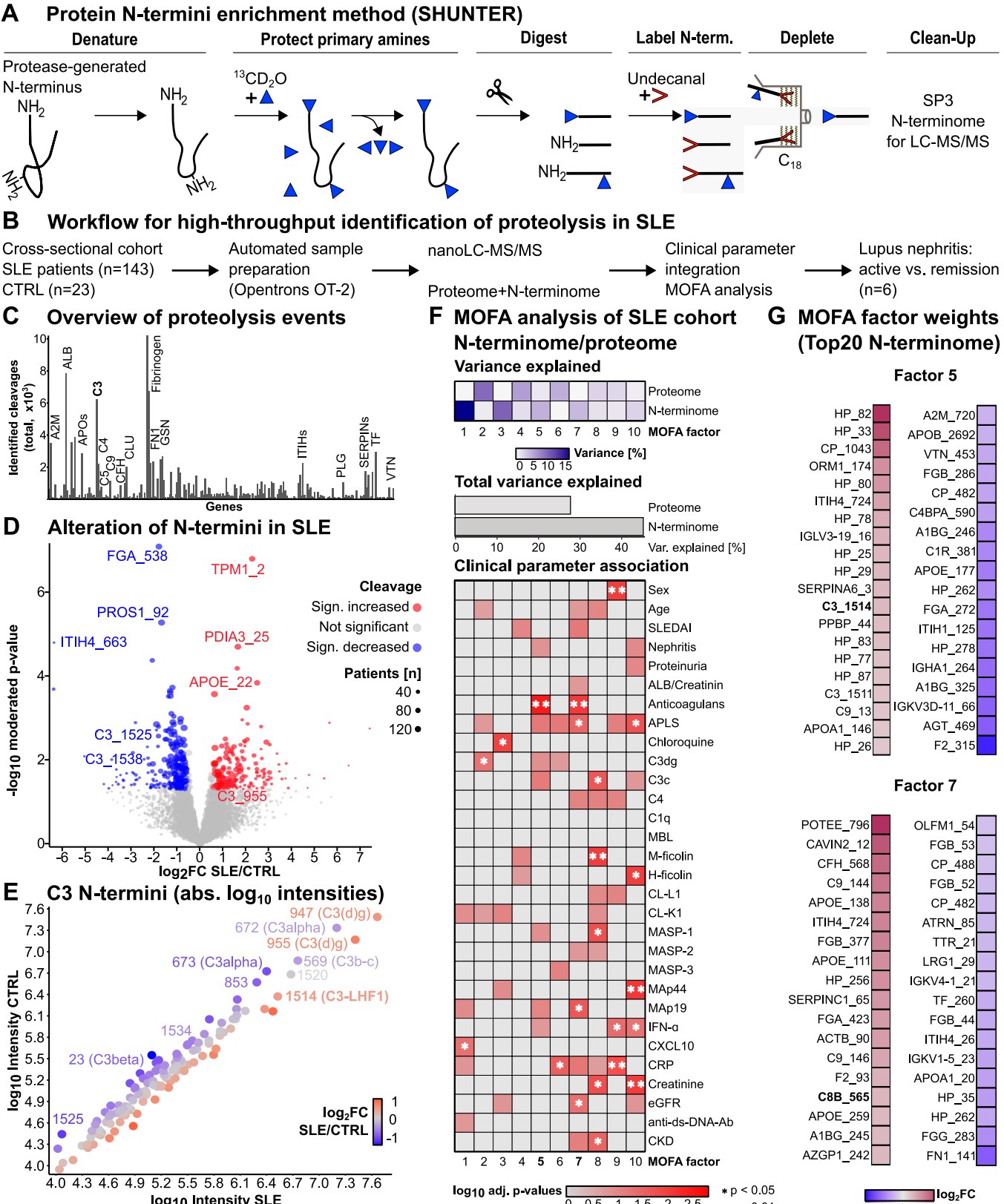

**A Protein N-termini enrichment method (SHUNTER)**

**B Workflow for high-throughput identification of proteolysis in SLE**

**C Overview of proteolysis events**

**D Alteration of N-termini in SLE**

**E C3 N-termini (abs. log$_{10}$ intensities)**

**F MOFA analysis of SLE cohort N-terminome/proteome**

**G MOFA factor weights (Top20 N-terminome)**

◄ **Figure 2.   Identifying human proteolytic patterns in systemic lupus erythematosus (SLE).**

(A) Overview of the applied N-terminome enrichment protocol, Simplex High-efficiency Undecanal-based N Termini EnRichment (SHUNTER). (B) Workflow for identification of SLE-specific inflammatory proteolysis. (C) Total identified cleavages in the human SLE cohort per protein. The histogram shows the product of cleavages in x number of samples. (D) Volcano plot demonstrating significant alteration of N-termini in SLE (increased/decreased cleavages are displayed in red and blue), focusing on key sites in the complement system. The point size encodes the number of identifications in individual samples. (E) Absolute $\log_{10}$ intensities for the identified C3 N-termini in the control and SLE samples span four orders of magnitude and identify two cleavages at the C-terminus of C3 (1520, 1514) as very abundant, as is the C3dg N-termini at 947 and 955. The data point color indicates the alteration in SLE/CTRL. (F) Multi-omics factor analysis (MOFA) of the SLE cohort demonstrates variance can mainly be explained by N-terminome in major factors 1, 3, 5, 7, 9 and 10. Clinical covariates are significantly associated with MOFA factors determining heterogeneity in SLE (* = adj. *p*-value < 0.05, *p*-value < 0.01, t-test of individual correlations). (G) Proteolytic cleavages driving N-terminome factors in SLE with a focus on anticoagulant treatment (factor 5) and eGFR (factor 7), sorted by the alteration in SLE versus CTRL. Source data are available online for this figure.

patient heterogeneity on a clinical level. N-terminomic data could explain most of the variance in the dataset, contributing almost exclusively to factors 1, 3, 5, 7, and 10 (Fig. 2F). Complete clinical data covering disease severity score, renal function, and biomarker profiles were integrated into the analysis. We found that N-terminome-driven MOFA factors were strongly associated with clinical covariates, such as CXCL10 cytokine (factor 1), chloroquine treatment (factor 3), anticoagulans (Factor 5), estimated glomerular filtration rate (eGFR, factor 7), and serum creatinine (Factor 10), all with significant interactions (adj. *p*-value < 0.05, Fig. 2F). Contributions to each MOFA factor were subsequently mapped (Dataset EV8), and detailed inspection of individual factors suggested that N-termini of complement proteins, particularly in the C-terminal part of C3 (pos. 1514, 1534) and C8b (pos. 565) contributed to the variance explained by the factors 3, 5 and 7 (Fig. 2F).

Several of the N-terminome driven MOFA factors associated with kidney phenotypes with a more relaxed statistical criterion, among these factors 5 (nephritis), 7 (eGFR, urinary albumin/ creatinine ratio), and 10 (proteinuria, nephritis, creatinine). To independently corroborate this association, we profiled serum from six patients with biopsy-proven lupus nephritis, a critical major organ manifestation of SLE, from a second site. Initial samples were obtained at the time of active disease. Later samples were obtained when patients, after induction therapy, had complete renal remission (CRR as per KDIGO (Rovin et al, 2024)). Consistent differences in the active versus remission stages were observed, particularly in N-termini of the complement system and for the C-terminal part of C3 (Fig. EV3E,F; Dataset EV9)—with the MOFA-prioritized C-terminal C3 fragments regulated during remission from lupus nephritis (Fig. EV3E). A complete repository of the top weights for all ten MOFA factors in the cross-sectional SLE cohort is depicted in Fig. EV4 (Dataset EV8).

## In vitro plasma signatures of complement proteases

To demonstrate which proteases could potentially generate the observed cleavages from the SLE cohort, we set up an in vitro protease substrate identification assay, adapting previously published workflows (Eckhard et al, 2016). We determined in vitro targets of four active proteases involved in the human complement system: MASP-1 for the lectin, MASP-3 for the alternative, and C1r and C1s for the classical pathway (Fig. 3A). All of these proteases were functional to cleave its preferred substrates (Morgan, 2000; Pihl et al, 2017, 2021), and inactive mutant proteases (Megyeri et al, 2013; Degn et al, 2014) were used as controls for MASP-1 and MASP-3. We incubated heat-treated plasma from four healthy

donors separately with active versions of the recombinant proteases —using inactive forms or no protease treatment as a control and subsequently performed quantitative N-terminome analysis. The protease-generated N-termini from total plasma were mapped, among these a fraction (7.5–10.4%) from complement proteins (Fig. 3B). All four applied proteases resulted in new fragments of C3 (Fig. EV2), with MASP-3 being the most active towards C3 (Figs. 3B,C and EV2B; Datasets EV10 and 11). C1s and C1r demonstrated several auto-digestion cleavages (Fig. EV2C,D). Sequence logos of all protease-generated motifs demonstrated target specificity and protease preference (Figs. 3D and EV2), generating the unbiased preferred cleavage motif of these important complement proteases for which only a handful (<6) substrates are found in protease databases like MEROPS (Rawlings et al, 2016) or TopFIND (Fortelny et al, 2015). Consistent with previous knowledge on substrates, MASP-1 and C1s cleaved after basic residues (K/R/H, Figs. 3D and EV2A,D). MASP-3 was also directed towards basic residues particularly in conjunction with acidic residues in proximity to the cleavage site (Figs. 3D and EV2B). C1r had a less clear target motif (Fig. EV2C). Observing cleavages at the C-terminus of human C3 within our cross-sectional SLE cohort (Fig. 2D,E) as well as in vitro assays using our selected four proteases (Fig. EV2), we determined whether these cleavages in C3 can be generated in an assay using purified human C3 and recombinant active MASP-1 and MASP-3 (Fig. 3E), compared to their inactive form (Morgan, 2000; Pihl et al, 2017, 2021). We found C-terminal processing of purified human C3 in the absence of plasma depending on the applied protease concentration, suggesting a putative generation of fragments mainly containing the C345c domain by active MASP proteases (Fig. 3E; Dataset EV12). We incorporated the cleavages from our in vitro assays in a network which revealed known but also potentially unknown protease-substrate relationships in human complement proteins (Fig. 3F). While in an in vitro system with known limitations (excess protease and/or use of heat-inactivated plasma), we demonstrate that proteolytic action within the complement system can be resolved in principle—recapitulating and expanding knowledge on potential protease substrates.

## Integration of N-terminomic signals across patients

We integrated N-terminome data from both in vitro and lupus plasma profiling as a circle plot, highlighting several relevant proteolytic fragments (Fig. 4A). The analysis covered virtually all the fragments described. Furthermore, several novel cleavage sites were observed consistently across different human datasets (Fig. 4A, outer tracks, Datasets EV7 and 9) and annotated as putative in vitro

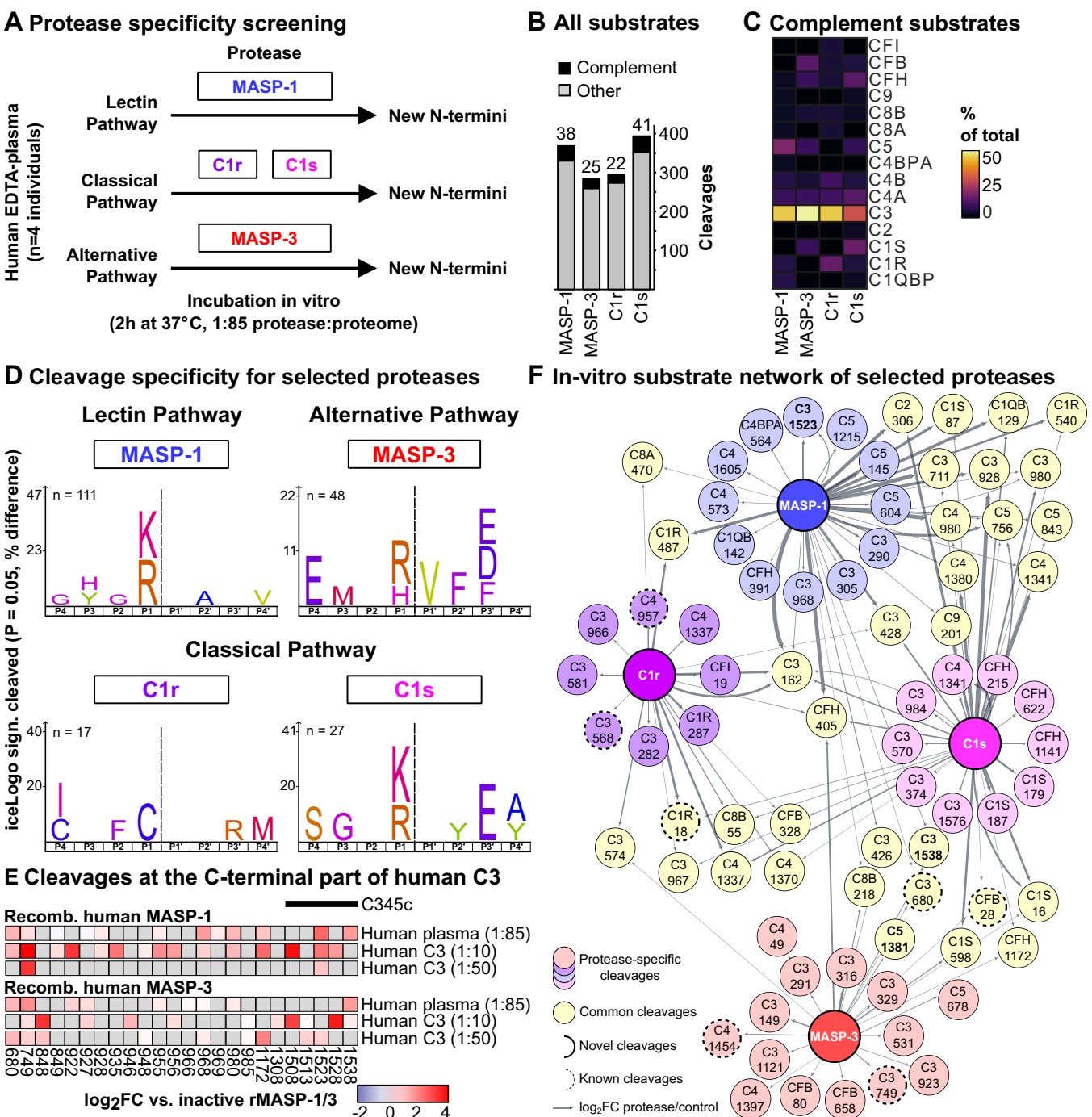

**Figure 3. Screening of specificity for selected proteases and putative substrates from the SLE cohort generated by the selected proteases.**

(A) Workflow for in vitro assays with human plasma ($n = 4$ individual donor samples): incubation of recombinant proteases with heat-inactivated human plasma (2 h at 37 °C, 1:85 protease:proteome ratio) to determine specific substrates for selected proteases of the complement system by subsequent SHUNTER N-termini enrichment. (B) Overview of the identified substrates for the individually evaluated proteases. (C) For all four proteases, cleavages mainly occur in complement component C3. (D) Cleavage motifs (iceLogos) for the selected four proteases derived from significantly cleaved protein N-termini (log2FC > 1 or novel cleavage & limma moderated t-test p-value < 0.05) following active protease treatment vs. respective control. (E) Overview of MASP-dependent cleavages at the C-terminus of human C3 in human plasma or recombinantly. Heatmap indicates log2FCs of N-termini with plasma matrix (Plasma + rMASP-1/3, $n = 4$ biological samples) as well as purified human C3 in the presence of active/inactive MASP-1 or MASP-3 (C3 + rMASP-1/3, $n = 3$ technical replicates). Abundant MASP-dependent cleavage at the C-terminal domain can be observed, giving rise to C3-LHF1 fragment. (F) Substrate network of significantly cleaved complement proteins by the investigated proteases. The number denotes the residue number. Source data are available online for this figure.

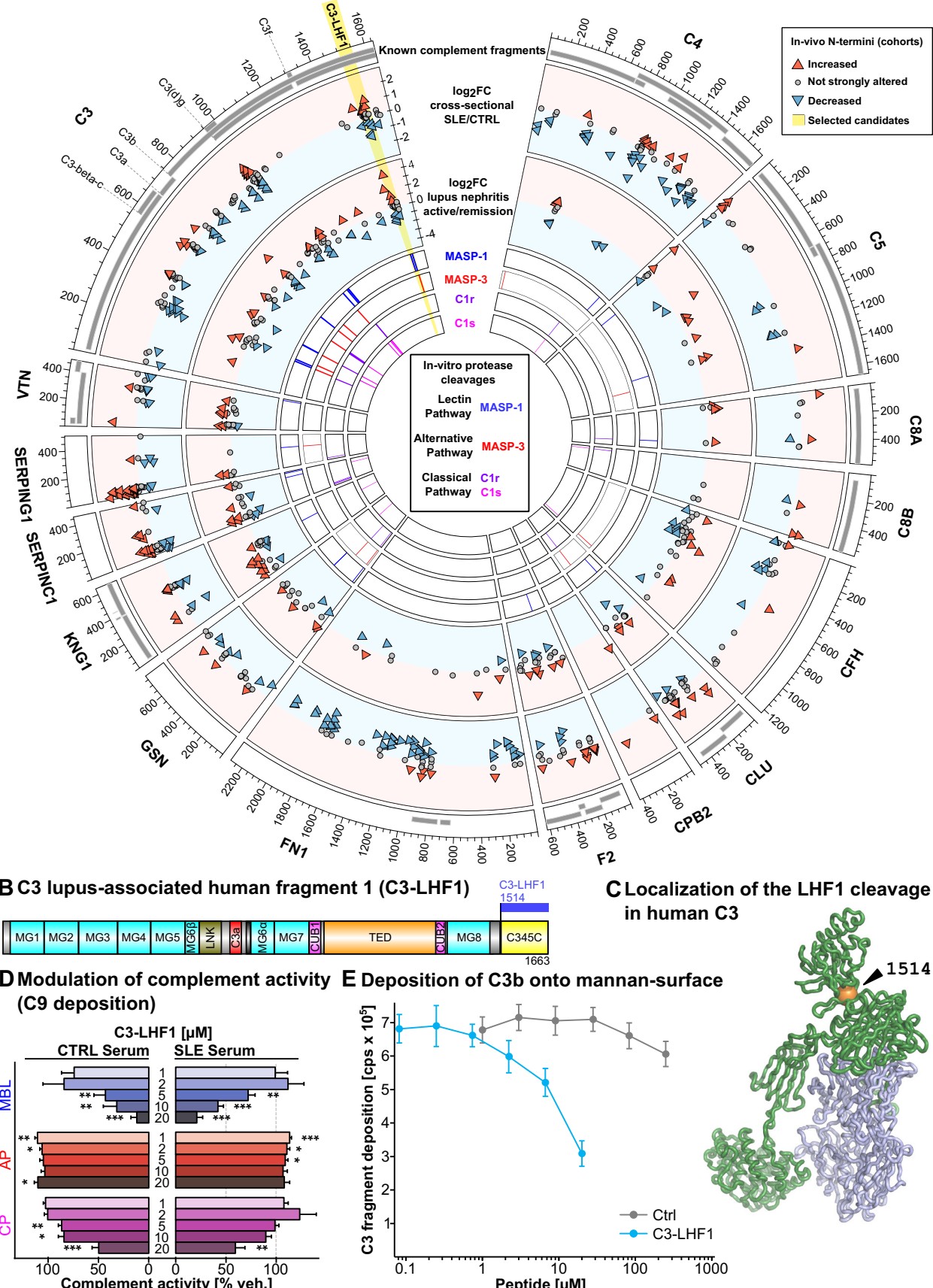

**A** Overview of the proteolytic processing landscape

**B** C3 lupus-associated human fragment 1 (C3-LHF1)

**C** Localization of the LHF1 cleavage in human C3

**D** Modulation of complement activity (C9 deposition)

**E** Deposition of C3b onto mannan-surface

**Figure 4.** **Integration of multi-layered proteolytic processing landscape in SLE.**

(A), Circle plot integrating known protein fragments in gray (layer one, outer layer), presence and regulation in cross-sectional SLE cohort (layer 2, cleavage in SLE), regulation in lupus nephritis patients (layer 3, cleavage in active vs. remission), and in vitro substrates (layer 4, 5, 6, 7), as well as total frequency of the N-termini identification in the center. Two MOFA-prioritized candidate N-termini are highlighted in yellow, potentially giving rise to novel fragments. (B) Postulated lupus human fragment (LHF) candidate, comprising the C-terminal part of C3, including the C345C domain (C3-LHF1). (C) Structural depiction of the C3-LHF1 fragment generating cleavage at position 1514 in human complement C3. (D) Recombinant C3-LHF1 inhibits complement activity in the classical and MBL pathways as measured by C9 deposition in reference to untreated vehicle control of the same patient ($n = 6$ for SLE/MBL, otherwise $n = 7$, mean ± SE, significance determined by paired t-test: *$p < 0.05$, **$p < 0.01$, ***$p < 0.001$; cf. source data for full list of p-values). (E) The effect of C3-LHF1 on C3b deposition was tested on mannan surfaces as a measure for MBL pathway activity ($n = 3$, mean ± SE). Source data are available online for this figure.

substrates of one of the four complement proteases studied (Fig. 4A, inner tracks, Datasets EV10 and 11). For complete data accessibility, we presented aggregated patient information data in a shiny application, providing a resource for circulating plasma proteolytic processing (proteolySee app: https://ahutz.shinyapps.io/proteolySee/). In the C3 protein, a putative fragment starting at position 1514 contains the C345c domain (Fig. 4B). This terminus is one of the most abundant in absolute intensity next to the classical termini raising C3(d)g, C3b-c, and C3a (Fig. 2E). This terminus is associated with factors 3 and 5 in the MOFA analysis, two factors associated with proteinuria and kidney function (FDR < 0.1 and 0.05, respectively, Fig. EV4A; Dataset EV8). Further visualization of the C3 structure revealed that this site, alongside all selected C3 sites with weight in the MOFA analysis, is accessible to proteolytic cleavage and not buried in the molecule (Figs. 4C and EV5).

## Complement inhibiting activity of novel complement fragments

Following these findings, we expressed a recombinant fragment termed C3-LHF1 (C3, lupus-associated human fragment 1), corresponding to residues 1514–1663 of the C-terminal region of human C3 (Fig. 4B). If cleaved here, a fragment will be generated that would lead to the release of the C345C domain (Fig. 4C). The C3-LHF1 fragment was produced in CHO cells with endotoxin removal and was sequence validated by mass spectrometry (Fig. EV6A). The secondary structure of the recombinant LHF-1 protein was analyzed using Synchrotron Radiation Circular Dichroism (SRCD, Fig. EV6B, for details please see Figure legend and methods), revealing the expected folding of the protein as compared to the PDB entry ID 2A73.

We next tested the impact of the novel complement fragment C3-LHF1 on the complement system activity in vitro by measuring the deposition of the MAC on relevant surfaces after activating the alternative, classical, or lectin pathways (Svar Wieslab assay). C3-LHF1 was highly active in suppressing the classical and MBL pathway, already at low micromolar concentrations (IC$_{50}$ of 5.4 µM for the MBL, 21 µM for the classical pathway on CTRL plasma), whereas the alternative pathway was not inhibited (Fig. 4D). Further analysis of iC3b deposition on mannan-surface revealed a decreased deposition by application of C3-LHF1 (Fig. 4E).

Since MASP-1 and MASP-3 could generate C-terminal peptides of C3 in vitro from a plasma matrix or in recombinant assays (Fig. 3E), we also tested if the C3-LHF1 fragment can, in principle, alter MASP proteolytic activity. To further understand the mechanism of action leading to complement self-control, we

performed analyses of the proteolytic action of MASP-1 and MASP-3 in a plasma matrix—now with and without presence of C3-LHF1, similar to our initial profiling of protease motifs (Fig. 3). Using a triplex stable isotope labeling approach (Fig. EV6C) and focusing on classical sites known to be propagated in the complement system, we found that in the presence of MASP-1 (Fig. EV6D; Dataset EV13) as well as MASP-3 (Fig. EV6F; Dataset EV14) induced the termini indicating C5 cleavage, but reduced the abundance of the C4 N-terminus at pos. 687, giving rise to the short-lived anaphylatoxin (Fig. EV6D,F). The C4 cleavage is also essential for forming the short-lived C4b2 C3-convertase—which is central for both the classical and MBL pathway. Addition of 20 µM C3-LHF1 to active MASP-1/-3 during the incubation was able to reverse the decrease of the N-terminus C4b at 687 (Fig. EV6E,G, N-terminus C4_687 marked in black).

## Stability of novel complement fragment C3-LHF1

A key question is whether a putative fragment comprising C3-LHF1 N-terminus would stabilize in circulation or if it is a transitional cleavage product. While high abundance (Fig. 2E) of the C3-LHF1 N-terminus suggests a certain degree of stability, it does not formally prove it. To determine stability in vivo, we performed half-life studies in mice—with the tradeoff of a non-identical complement system to humans. We performed pulsed stable isotope labeling (dietary $^{13}$C$_6$-Lysine) in wild-type mice for two weeks (Krüger et al, 2008; Rinschen et al, 2018, 2022). Here, the relative $^{13}$C-isotope signal in the serum of stable isotope labeled mice indicates if fragments are derived from a freshly synthesized protein (high $^{13}$C isotope signal) or a protein that is very stable in circulation (low $^{13}$C isotope signal, Fig. 5A). It is important to acknowledge that these data do not measure true protein half-lives, as synthesis and degradation are intertwined—disentangling them would likely require triple labeling (Boisvert et al, 2012). We subjected serum to N-termini enrichment and were able to detect a labeled and a non-labeled version of each N-terminus. As expected, the isotope labeling ratio in the serum increased from week 1 to week 2 (Fig. 5B). Interestingly, termini-dependent differences in the relative incorporation (log$_2$ ratio of $^{13}$C$_6$ labeled lysine/control lysine) were found, particularly for proteins in the complement system (Fig. 5C; full overview in Appendix Fig. S2, Dataset EV15). Interestingly, the C3 N-termini starting at residue E1515 and F1519 (Fig. 5C), very close to the orthologues (A1514/I1520) discovered in humans (Figs. 2E and 4A, sequence alignment in Fig. 5D) are associated with low $^{13}$C isotope content and thus, likely a longer half-life of the protein fragment in circulation.

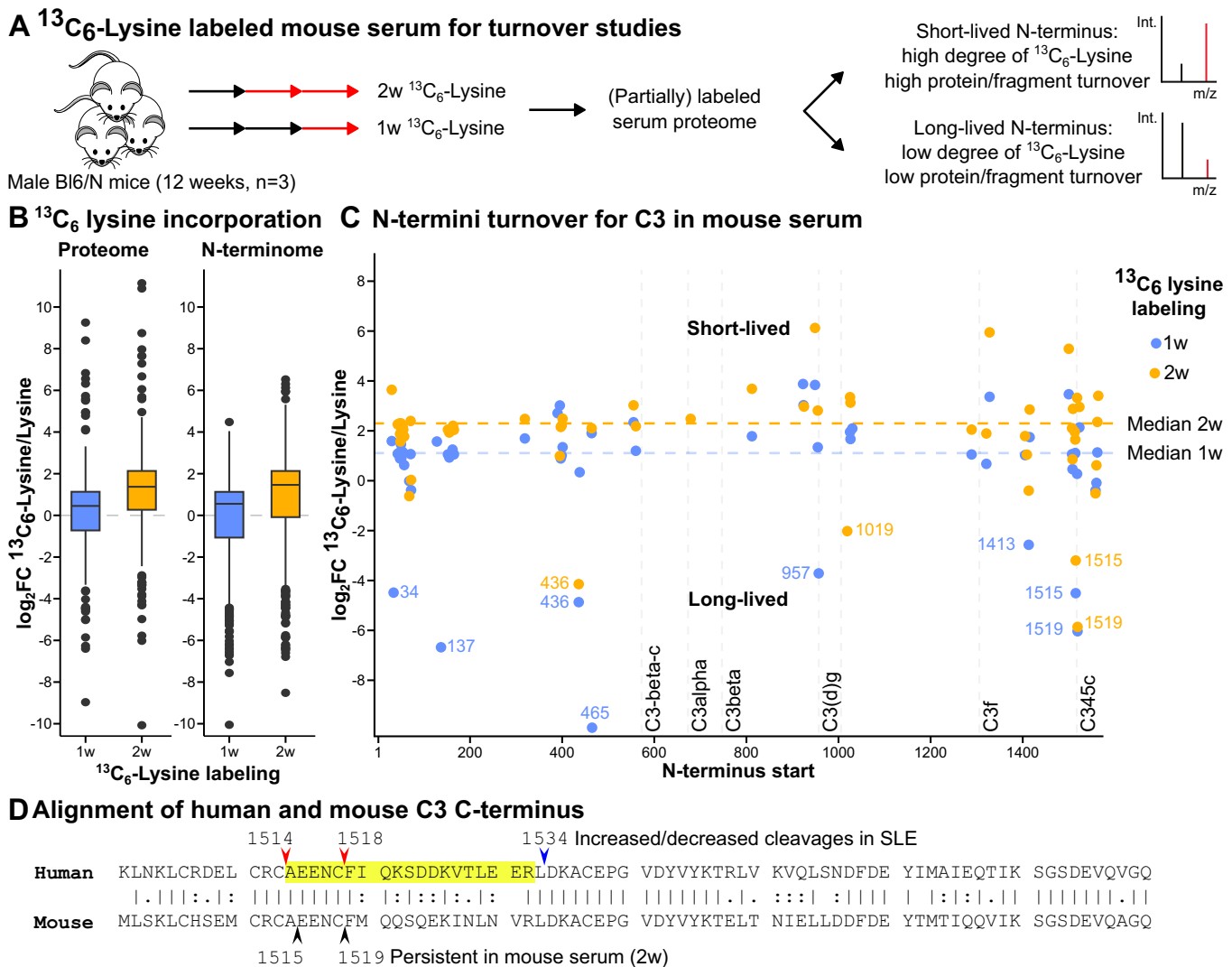

**Figure 5. Lifetime of mouse complement fragments based on pulsed stable ¹³C isotope labeling.**

(A) Mice were fed with $^{13}C_6$-lysine stable isotope labeled diet for two weeks. The plasma proteome was partially labeled over time, and purification of N-terminal peptides was performed after 1 or 2 weeks of labeled diet. (B) Comparison of isotope labeling in proteome and N-terminome of the mouse serum after 1 week (blue) and 2 weeks (orange) of isotope labeling ($n = 3$ biological replicates; boxplot line represents the median, whereas the box covers the first and third quartiles and the whiskers extend to the 1.5x interquartile range from the box). (C) Isotope-labeling ratio of C3-N-termini highlights long-lived, stable cleavages, especially in the C-terminal region of mouse C3 (P1413, E1515, F1519). (D) Alignment of the C-terminal region of human and mouse C3 depicting the identified/regulated cleavages in the cross-sectional SLE cohort (red/blue arrows) and the persistent cleavages after two weeks of isotope labeling in mouse serum (black arrows). The identified candidate C3-LHF1 is highlighted in yellow and corresponds to a region of stable C3 N-termini in the isotope labeled mouse serum.

## Configuration of circulating novel complement fragment C3-LHF1

A long-range disulfide bridge links cysteines 873 and 1513 in C3 (Dolmer and Sottrup-Jensen, 1993). Structural models of C3 and C3b suggest that a cleavage at residue 1514 would release a free C-terminal fragment. In contrast, if cleavage only occurs at residue 1520 or 1534, the C-terminal fragment may remain with the rest of C3 through the cysteine 1518–1590 disulfide bridge unless the disulfide bridge is reduced. To check if the fragment is still attached to or in complex with the rest of C3 in humans, we analyzed a single human SLE plasma sample (patient with lupus nephritis and

a severe SLEDAI disease score of 10) fractionated by size exclusion chromatography (SEC) combined with N-terminome and proteome analysis of the corresponding fractions (Fig. EV7A,B). The analysis revealed that the N-termini C3_1514, C3_1520, and C3_1534 were distinct from the main form of C3 and eluted at a later retention time (Fig. EV7C; Dataset EV16), consistent with a potential free form in plasma or putatively linked to C3c (proteolytic processing in C3 is schematically given in Fig. EV7D). We also performed interactomic analysis in human plasma, using C3-LHF1 and C3 as purified by a C3-specific nanobody (hC3Nb2, Pedersen et al, 2020b) as a control ($n = 4$ individual plasma samples for healthy controls and SLE patients, Appendix Fig. S3A,

Dataset EV17). To compensate for any alterations in the bulk protein abundance among CTRL and SLE, we also investigated the global plasma proteome in those samples (Appendix Fig. S3B). While our data found a classical engagement of C3 with C3 and C4, nidogen-1 (NID1), and plasminogen (PLG), C3-LHF1 also engaged with other lupus- and autoimmune-related proteins, such as WDR1, RPL22, and PXDN (Appendix Fig. S3C), supporting its mechanism of interacting with surface binders.

## Bioactivity of C3-LHF1

Next, we decided to analyze if the discovered fragments have further bioactivity that could explain their linkage to lupus phenotypes. First, we used a cell painting assay (Bray et al, 2016; Svenningsen and Poulsen, 2019) to check if the compound was bioactive. The cell painting assay is a five-plex staining method followed by an automated morphological analysis that can uncover the bioactivities of molecules based on the distinct repositioning of organelle features (Appendix Fig. S4A, Bray et al, 2016). We used the human osteosarcoma U-2OS cancer cell line for assessing changes in cell morphology for C3-LHF1 (Appendix Fig. S4B). Subsequent data analysis of the cell painting profiles displayed a bioactivity at low micromolar concentrations of 5–10 μM for C3-LHF1 (Appendix Fig. S4C), but no decreased viability in the assayed concentration ranges for C3-LHF1 (1–20 μM). Subsequently, we investigated bioactivity in native human granulocytes by analyzing the effect of C3-LHF1 on surface CD62L shedding (Kishimoto et al, 1989), revealing an activation of the immune cells ($n = 5$ individual donors, Appendix Fig. S4D).

## C3-LHF1 interacts and activates human gp130 (IL6ST)

The clinical linkage of C3-LHF1 to renal outcomes, as well as its bioactivity, was fascinating to us. To understand potential targets of C3-LHF1 in human kidneys, we performed pull-down experiments using C3-LHF1 as bait to investigate its interaction with signaling components. We incubated C3-LHF1 bound to beads with human kidney lysates and used unbiased mass spectrometry-based detection to identify interacting proteins. Both vehicle and denatured C3-LHF1 protein (carbamidomethylated and heat-denatured; C3-LHF1$^{DN}$) were used as controls (Fig. 6A; Dataset EV18). Interestingly, C3-LHF1 was found to bind to IL6ST (gp130), a key mediator of IL6 signaling, particularly trans-signaling (Rose-John et al, 2023), in renal tissue (Fig. 6B). To establish any potential interacting proteins in an independent second, unbiased assay, we performed thermal proteome profiling – proteome integral solubility alteration assay (TPP-PISA, Batth et al, 2024) with 1 or 10 μM C3-LHF1 on human kidney tissue (Fig. 6C; Dataset EV19). Proteins interacting with C3-LHF1 will be stabilized and confer more resistance to heat-denaturation as applied during the TPP-PISA workflow (5 min at 53/56/59 °C). Addition of the C3-LHF1 fragment significantly stabilized a considerable number of proteins (Fig. 6C, marked in black), among those IL6ST, which represented the most stabilized protein (log$_2$FC of +1.32 for 1 μM C3-LHF1) among the significant C3-LHF1 pull-down candidates from human kidney (Fig. 6B). We thus decided to analyze the effects on IL6 signaling.

Using IL6Rα/IL6ST cell reporter assays, we demonstrated that C3-LHF1 dose-dependently engaged the IL6 receptor complex,

consisting of IL6ST and IL6Rα—and leading to reporter activation, which depends on engagement of STAT3 (Fig. 6D). With descending concentrations of IL6, partial inhibition of IL6 signaling was observed. Activation was seen in the presence of low IL6 concentrations (Fig. 6E). These findings suggest that C3-LHF1 acts as a partial agonist on IL6ST (gp130). Furthermore, the presence of Tocilizumab, an antibody and IL6Rα inhibitor, did not prevent C3-LHF1 from eliciting a response, indicating that C3-LHF1 may signal independently of IL6Rα (Fig. 6F), and directly activate IL6ST. Directly targeting C3-LHF1 with a nanobody specific for the C-terminal C345c domain of complement C3 (hC3Nb3, Pedersen et al, 2020a), we could demonstrate the lack of a IL6Rα/IL6ST reporter response (Appendix Fig. S5A). The IL6Rα/IL6ST reporter response is also missing if we apply carbamidomethylated and heat-denatured C3-LHF1 (C3-LHF1$^{DN}$) instead of the native C3-LHF1 (Appendix Fig. S5B).

## C3-LHF1 activates IL6ST and JAK-STAT signaling in kidney organoids

Several lines of evidence demonstrated the relevance of C3-LHF1 for effects on the kidney system (Fig. 7A). We used our ipSC-derived human kidney organoid system, in which molecular signatures can be observed that are similar to those in patients with inflammatory kidney disease (Lassé et al, 2023). We analyzed the proteome of kidney organoids at a depth of 10582 identified proteins (Dataset EV20). In baseline conditions, we found that C3-LHF1 (18 μM for 48 h) increased JAK3 and STAT3, a signaling axis that is downstream of IL6ST (gp130)—both being significantly regulated proteins (Fig. 7B). We also treated kidney organoids with C3-LHF1 in the presence and absence of TNFα, a signaling molecule associated with progression of lupus pathologies (Fig. 7C). Again, an increase in JAK3 and STAT3 was observed. Plotting of the C3-LHF1 perturbation in the organoid proteome revealed a relative increase in STING1, a protein involved in coordinating proinflammatory responses (Fig. 7D). To test if this was functional, we tested if C3-LHF1 induced CXCL10, a known target gene of STING1 that also emerged as a key determinant of degradome variability in our initial multi-omics factor analysis (Fig. 2F). CXCL10 secretion in the kidney organoid supernatant was significantly increased by C3-LHF1 application (Fig. 7E, 27.7% increase with 18 μM C3-LHF1 after 48 h compared to PBS vehicle control; two-sided t-test $p$-value = 0.02). To analyze signaling further, we also performed phosphoproteomics (Dataset EV21) at a depth of 72874 phosphosites, among which 9789 were significantly altered ($|\log_2 FC| > 0.58$ & BH-adj. t-test $p$-value < 0.05). Phosphorylation abundance demonstrated that IL6ST was the most differentially phosphorylated receptor, demonstrating increased phosphorylation at the activating TBK S/T sites (Kim et al, 2013), as well as the autophosphorylation site Y814 (Fig. 7F) that has recently been linked to decreased tissue regeneration (Shkhyan et al, 2023). Focusing on already annotated tyrosine phosphorylation as a central measure to regulate signaling pathways, we could identify Y705 in STAT3 as the most significantly up-regulated tyrosine phosphosite upon C3-LHF1 application in presence of TNFα (log$_2$ ratio +2.84, Fig. 7G). Interestingly, nephrin tyrosine phosphorylation was increased at an activating site (Verma et al, 2006), that we have also shown to be increased in proteinuric disease upon selective damage to nephrin-expressing podocyte cells

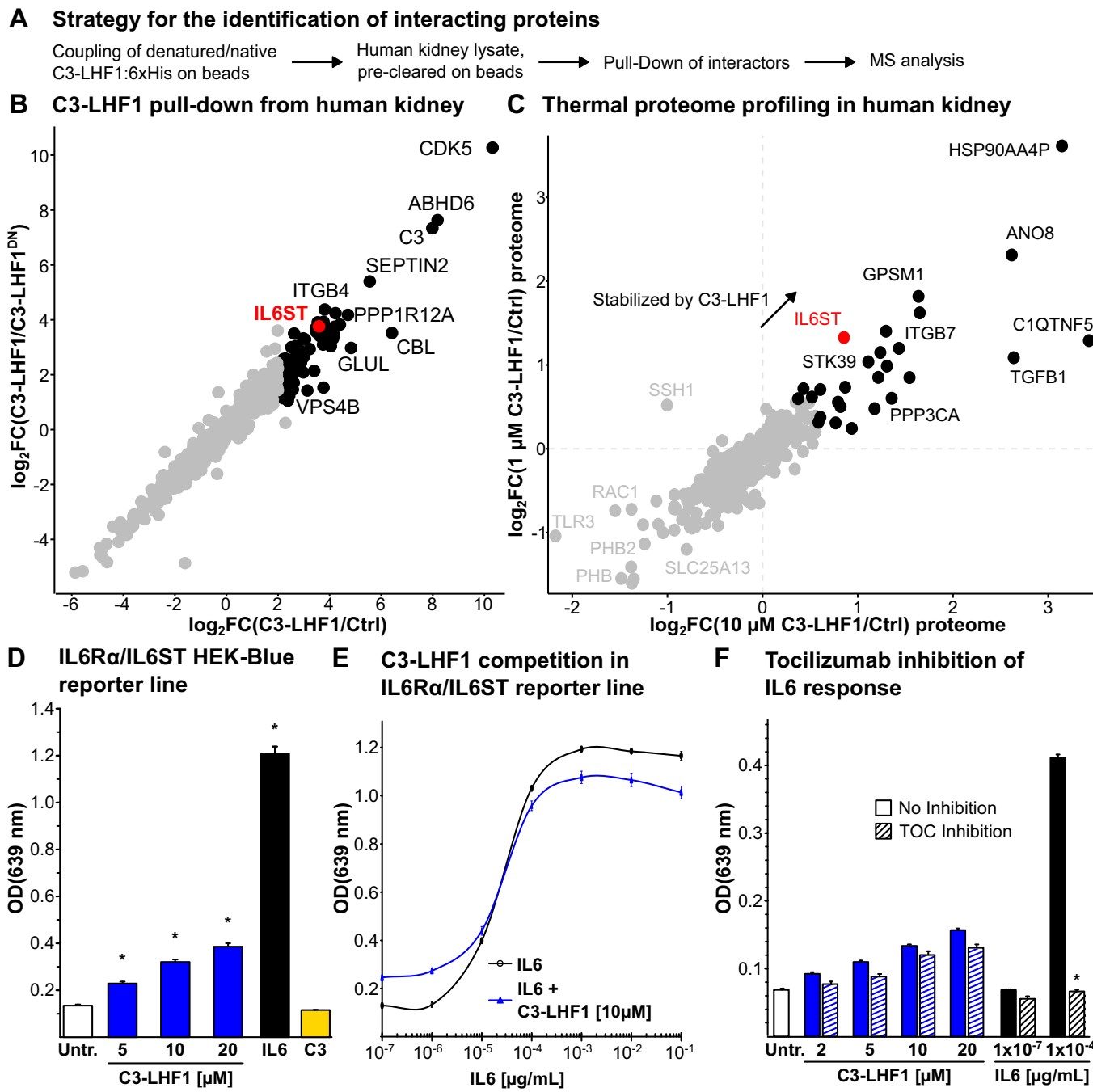

**Figure 6. Characterization of the bioactivity of the C3 fragment C3-LHF1.**

(A) Pull-downs with native and denatured (heat-inactivated and carbamidomethylated) C-terminally His-tagged C3-LHF1 were performed on human kidney homogenate. (B) Significantly enriched (BH-adj. t-test $p$-value < 0.05, $\log_2$FC > 1) proteins in the kidney pull-downs included the IL6 signaling component IL6ST (gp130). (C) Thermal proteome profiling by proteome integral solubility alteration assay (TPP-PISA) in human kidney lysates to identify putative C3-LHF1 interacting proteins ($n = 3$ technical replicates). Only sign. altered (BH-adj. t-test $p$-value < 0.05, $\log_2$FC > 0.58) and plasma membrane-resident proteins are displayed after addition of 1 or 10 µM C3-LHF1 (1 h, 37 °C). Proteins stabilized by interactions with C3-LHF1 display a higher abundance in the treated samples (IL6ST depicted in red). (D) C3-LHF1 effect on IL6R/IL6ST signaling was tested in a HEK-Blue IL6Rα/IL6ST reporter line (incubation o/N at 37 °C). C3-LHF1 effect was concentration dependent, and activating, unlike bulk C3 ($n = 4$, mean ± SE; two-sided t-test $p$-value < 0.001 = *; IL6 used at $3 \times 10^{-6}$ µg/mL, $p$-value vs. untreated control: 0.0000001; $p$-values for 5-10-20 µM C3-LHF1 vs. untreated control: 0.000119, 0.000009, and 0.000006, respectively). (E) In HEK-Blue IL6Rα/IL6ST cell line with both IL6 and C3-LHF1 (o/N at 37 °C), a partial competition can be observed at IL6 concentrations of >1 × 10⁻⁴ µg/mL ($n = 4$, mean ± SE). (F) Tocilizumab (TOC, 2 µg/mL for 3 h at 37 °C) did not inhibit the C3-LHF1 response in the IL6Rα/IL6ST cell line, unlike the IL6 response at 1 × 10⁻⁴ µg/mL ($n = 4$, mean ± SE, two-sided t-test $p$-value < 0.001 = *; $p$-value IL6 + tocilizumab vs. IL6 at 1 × 10⁻⁴ µg/mL = 1.5 × 10⁻⁹). Source data are available online for this figure.

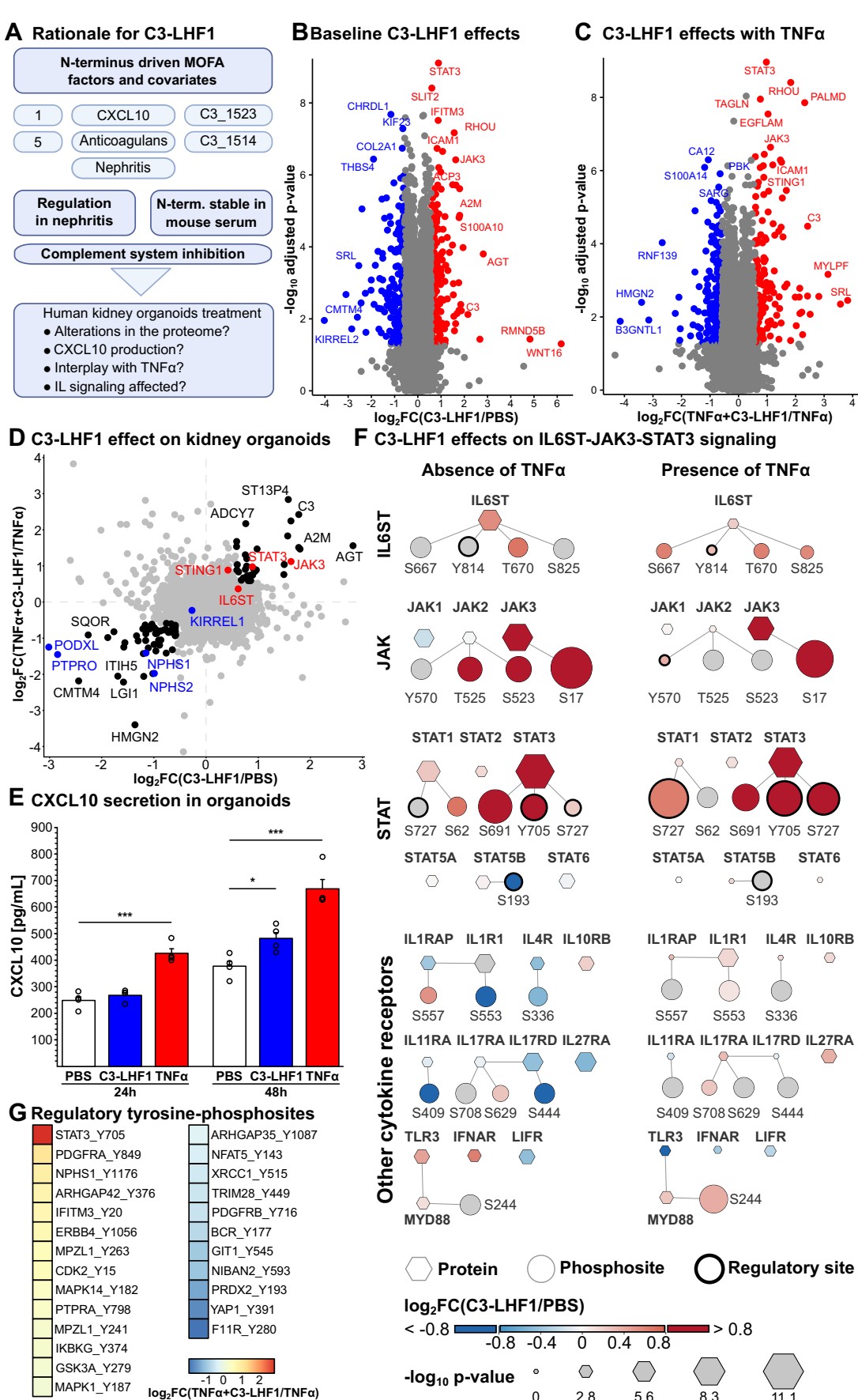

**A** Rationale for C3-LHF1

**B** Baseline C3-LHF1 effects

**C** C3-LHF1 effects with TNFα

**D** C3-LHF1 effect on kidney organoids

**E** CXCL10 secretion in organoids

**F** C3-LHF1 effects on IL6ST-JAK3-STAT3 signaling

**G** Regulatory tyrosine-phosphosites

**Figure 7. Innate immune signaling activity of C3-LHF1 on human kidney organoid and cell systems.**

(A) Rationale for assessing the activity of C3-LHF1 on kidney organoids. The various elements are discussed in the main text. (B) Baseline alterations in the kidney organoid proteome upon application of 18 μM C3-LHF1 for 48 h (significance threshold: BH-adj. t-test $p$-value < 0.05, $|log2FC| > 0.58$, $n = 4$ technical replicates) displays reduction of CMTM4 and an increase of JAK3/STAT3. (C) Proteome alterations upon TNFα (25 ng/mL) treatment and combined TNFα (25 ng/mL) and C3-LHF1 (18 μM) treatment for 48 h (sign. threshold: adj. $p$-value < 0.05, $|log2FC| > 0.58$, $n = 4$ technical replicates). (D) Combined analysis of C3-LHF1/PBS and C3-LHF1 + TNFα/TNFα reveals significantly C3-LHF1 affected proteins (BH-adj. t-test $p$-value < 0.05): podocyte markers are down-regulated by C3-LHF1 application (KIRREL1, NPHS1, NPHS2, PTPRO, PODXL—marked in blue), while the inflammatory response by the IL6ST-JAK3-STAT3 axis is up-regulated (STING1, IL6ST, JAK3, STAT3—marked in red). (E) CXCL10 secretion as a measure of inflammatory response was assayed in human kidney organoid supernatants and revealed an increase in CXCL10 secretion after 48 h of C3-LHF1 incubation ($n = 4$ technical replicates, mean ± SE, two-sided t-test significance indicators: *$p$ < 0.05, ***$p$ < 0.001; $p$-value for TNFα vs. PBS, 24 h = $3.6 \times 10^{-4}$, C3-LHF1 vs. PBS, 48 h = $1.9 \times 10^{-2}$, and TNFα vs. PBS, 48 h = $6.7 \times 10^{-4}$). (F) Combined proteomic and phosphoproteomic analysis ($n = 4$) reveals significant changes in the IL6ST-JAK3-STAT3 signaling (t-test, BH-adj. $p$-value < 0.05) axis upon C3-LHF1 application in the absence and presence of TNFα. (G) Regulatory tyrosine (Y) phosphosites in the human kidney organoid phosphoproteome upon combined C3-LHF1 + TNFα treatment for 48 h. STAT3 phosphorylation at Y705 is the main induced phosphosite by C3-LHF1 in the presence of TNFα. Source data are available online for this figure.

(Hengel et al, 2024). Consistent with a detrimental effect of C3-LHF1, we observed even further decrease of podocyte-specific markers in organoids (NPHS1, nephrin, NPHS2, podocin, PODXL, podocalyxin, PTPRO tyrosine kinase) as top-regulated proteins (Fig. 7D).

## Discussion

Inhibitors of the human complement system can potentially alleviate several debilitating diseases, including autoimmune, rheumatic, eye, and kidney diseases. Many complement inhibitors are on the market or under development (West et al, 2024), including anti-C5 antibodies (Legendre et al, 2013), or complement receptor blockers (Jayne et al, 2021). Particularly, inhibitors of proteolysis are emerging as treatments for C1s in cold agglutinin disease (CAD, sutimlimab, Mullard, 2022; Broome, 2023), C3 (APL-2, compstatin, pegcetacoplan, Hillmen et al, 2021; Lamers et al, 2022; Mollnes et al, 2022), factor B and D for paroxysmal nocturnal hemoglobinuria (PNH, danicopan & iptacopan, Kang, 2024; Latour et al, 2024; Perkovic et al, 2025) or MASP-2 in thrombotic microangiopathy (TMA, narsoplimab, Elhadad et al, 2020). Recently, specific complement fragments have been targeted, such as vilobelimab for C5a in antineutrophil cytoplasmic antibody (ANCA) vasculitis, an antibody currently tested in patients resistant to complement inhibitor therapy (Petr and Thurman, 2023), suggesting that knowledge on immunoactivity of complement fragments can be clinically exploited.

Our findings expand the understanding of complement-mediated proteolysis by revealing previously uncharacterized complement fragments with functional and clinical significance. Contrary to a view of a linear complement activation cascade, our result highlights the complex network of cleavage events that modulate diverse protein targets and pathways. The use of high-sensitivity mass spectrometry, which surpasses traditional methods like gel-based Edman sequencing, enabled the identification of smaller but clinically relevant fragments such as C3-LHF1 (~150 amino acids). Among the previously uncharacterized cleavages, those at A1514 and I1520 were especially abundant, with A1514 showing distinct regulation in lupus. This fragment ranks in abundance directly after the well-known C3 fragments C3b, C3a, and C3dg (Fig. 2E)—in 5th place of all C3 cleavages. The proteolytic sites 1514 and 1534 in the C345c domain are present in C3, C3b, iC3b, and C3c—these sites are accessible in all known

C3, C3b and C3c structures (Fredslund et al, 2006; Janssen et al, 2005, 2006), which facilitates protease access. For other prioritized fragments, structural information helps to clarify how proteolysis could occur. For instance, the two C3 cleavage sites at D961 and L982 are in the CUB domain (Fig. 4B, purple). Structurally, D961 is accessible in all functional states of C3, whereas L982 is buried in both native C3 and C3b, and the surrounding residues are not highly accessible. However, both residues are accessible in iC3b. The C3dg fragment 955-1303 is released when C3b is inactivated by complement Factor I (Fig. EV7D). In iC3b and C3dg, the region containing D961 and L982 are disordered. These new cleavages are, therefore, likely to be derived from iC3b rather than directly from C3 or C3b.

Complement self-regulation was discovered as a process orchestrated by the C3-LHF1 fragment. The cleavage at C3 A1514 releases C3-LHF1, containing the C345c domain: The orthologue fragment is stable in mice serum, evidenced by whole-body isotope incorporation studies (Fig. 5). In human plasma, the C3-LHF1 domain interacts with DNA binding proteins, but also other complement factors, particularly C4 (Appendix Fig. S3C). Functionally, C3-LHF1 effectively attenuated classical and, more prominent, the lectin pathway-dependent C9 deposition while leaving the alternative pathway largely unaffected (Fig. 4D). A possible explanation is that the C3-LHF1 fragment interferes with MASP-1/MASP-2 function and the formation/activity of the C4b2a CP C3 convertase, since presence of C3-LHF1 suppresses C3b deposition on mannan-surfaces (Fig. 4E). Correspondingly, we observed that C3-LHF1 reversed the MASP-1-and MASP-3-induced reduction of the C4_687 terminus in vitro (Fig. EV6E,G). This suggests an early entry point for potential complement self-control upstream in the cascade. But C3-LHF1 also appears to interfere with the activity of the CP/LP C5 convertase (Fig. 4D) in line with the fact that the hC3Nb3 nanobody binding to the C3 C345c domain is a partial antagonist for the CP/LP C5 convertase (Pedersen et al, 2020b).

Equally, the capacity of C3-LHF1 to modulate IL6ST/gp130 reveals novel connections between the innate immune system and adaptive cytokine signaling. Unbiased studies—affinity purification of the protein from human kidney tissue lysate—demonstrated that C3-LHF1 can bind to IL6ST (gp130) and act as a partial agonist on this receptor. This appears to occur independent of IL6Rα function (a receptor targeted in many rheumatic diseases). IL6ST (gp130) is a ubiquitously expressed protein that is required as the adapter for all IL6 family cytokines: IL-6, IL-11, IL-27, LIF, OSM, IL-35,

cardiotrophin-1, and others (Rose-John, 2018). Genetically, dominant negative mutations in IL6ST are linked to hyper-IgE syndrome (Béziat et al, 2020). This is interesting because autoreactive IgE is discussed as a potential target (Hasni et al, 2019) or at least mechanistic biomarker in SLE and lupus nephritis (Henault et al, 2016; Himbert et al, 2024). Consistent with IL6ST activation (Béziat et al, 2020), the C3-LHF1 fragment induced a strong IL6ST autophosphorylation on tyrosine and phosphorylation at TBK sites and a dominant STAT3 pathway activation—shown in human kidney organoids (Fig. 7F,G). Considering the entire phosphoproteome and proteome of these organoids, JAK3-STAT3, on a global level, is the leading signal transduced by C3-LHF1, even in the presence of another potent cytokine such as TNFα (Fig. 7F). Notably, this occurs together with an increased nephrin phosphorylation (Fig. 7G) which can also be observed in humans when immune complexes bind to nephrin, causing proteinuria (Hengel et al, 2024)—and a dominant reduction of podocyte marker proteins (Fig. 7D). A strongly decreased protein by C3-LHF1 was CMTM4 (Fig. 7D, depicted in blue), an essential part of the IL17 receptor responsible for limiting tissue damage in autoimmune diseases (Mezzadra et al, 2017; Knizkova et al, 2022), suggesting that the response is more fine-tuned than just detrimental. Further links of C3-LHF1 to Lupus and particularly lupus nephritis also come with its physical association in plasma to peroxidasin (Appendix Fig. S3C), a protein suggested to be an autoepitope in lupus nephritis (Manral et al, 2019), or lupus-linked proteins such as RPL22 (Kegerreis et al, 2019), and WDR1 (Kile et al, 2007). Together, this positions C3-LHF1 as an immunoactive mediator rather than a mere byproduct of complement activation—capable of shaping immune and inflammatory processes independent of C3.

While these findings broaden our understanding of complement-related proteolysis, several questions remain. First, more functional mining of the proteolytic landscape—beyond the complement system—needs to be performed carefully but offers clues to study the still enigmatic links between complement, coagulation, and fibrosis (Stark et al, 2024). For this, we have provided aggregated data in an online resource (available through GitHub or directly at https://ahutz.shinyapps.io/proteolySee/, Appendix Fig. S6). A prime candidate for functional studies, for instance, is a site in C5 at I1381, which not only can be explained by eculizumab binding (Fig. EV1F,G) but also is a strong determinant of lupus patient heterogeneity and CXCL10 levels (Figs. 4A and EV4A, Factor 1). Second, the upstream proteases responsible for driving the proteolytic events in vivo have yet to be determined. However, our first determination of preferred sequence motifs for known proteases in a plasma matrix offers tools and starting points for future mechanistic work. For C3-LHF1, we have determined that MASP-1 and MASP-3 proteases can generate C3-LHF1 N-termini in this part of the protein in vitro. Third, the role of patient-specific or disease-specific cofactors (e.g., autoantibodies, microbial triggers) in shaping the complement fragment repertoire requires deeper investigation. The multi-omics factor analysis, however, links C3-LHF1 to the presence of kidney phenotypes, a finding further corroborated through subsequent experiments. With all limitations in mind, it should be stated that proteolytic information is not covered by "next generation" affinity-based serum proteomics analyses such as OLINK and SOMASCAN.

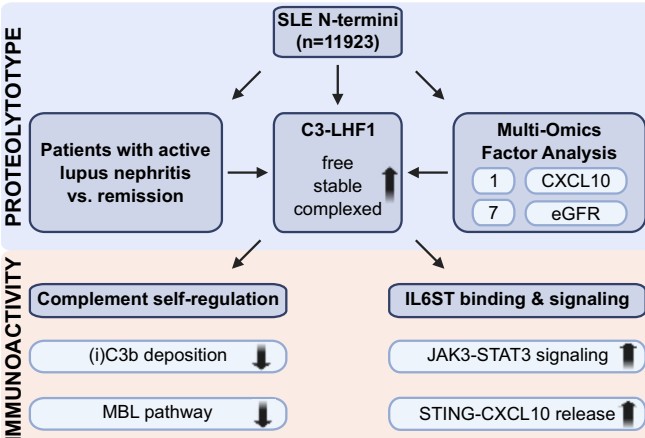

**Figure 8.** Summary of our study linking proteolytotype to immunoactivity, exemplified by proteomic effects of C3-LHF1.

In conclusion, this study uncovers a new dimension of immune crosstalk: a proteolytic plasma network that generates immunoactive fragments with intrinsic self-regulatory capabilities to modulate inflammation and tissue injury in complex disease contexts (Fig. 8). Deciphering these immunoactivities may pave the way for innovative diagnostic and therapeutic strategies in complement-driven pathologies, potentially benefiting a wide range of inflammatory and autoimmune diseases.

## Methods

**Reagents and tools table**

| Reagent/Resource | Reference or Source | Identifier or Catalog Number |
|---|---|---|
| **Experimental models** | N/A | |
| Human kidney organoids | (Lassé et al, 2023) | |
| HEK-Blue IL6 reporter line | Invivogen | #hkb-hil6 |
| U-2OS | ATCC | #HTB-96 |
| **Recombinant DNA** | N/A | |
| **Antibodies** | N/A | |
| Monoclonal IgG anti-human CD62L | BD Pharmingen | Clone: DREG-56, #559772 |
| Biotinylated anti-hC3d | Dako | #A0063 |
| hC3Nb2 anti-C3 nanobody | (Pedersen et al, 2020b) | |
| hC3Nb3 anti-C345c nanobody | (Pedersen et al, 2020a) | |
| **Oligonucleotides and other sequence-based reagents** | | |
| C3-LHF recombinant protein | C3:LHF1 sequence presented | Identified in Material and Methods MGWSCIILFLVATATGVHSAEENC FIQKSDDKVTLEERL-DKACEPGVDY VYKTRLVKVQLSNDFDEYIMAIEQTI KSGSDEVQVGQQRTFISPIKCREALK LEEKKHYLMWGLSSDFWGEKPNL SYIIGKDTWVVEHWPEEDECQDEE NQKQCQDLGAFTESMVVFGCP NHHHHHH |

| Reagent/Resource | Reference or Source | Identifier or Catalog Number |
|---|---|---|
| **Chemicals, enzymes and other reagents** | | |
| Sera-Mag SpeedBeads Carboxyl Magnetic Beads, hydrophilic | Cytiva | #45152105050250 |
| Sera-Mag SpeedBeads Carboxyl Magnetic Beads, hydrophobic | Cytiva | #65152105050250 |
| cOmplete EDTA-free protease inhibitor cocktail | Roche | #05056489001 |
| $^{12}CH_2O$ formaldehyde | Sigma-Aldrich | #252549 |
| $^{13}CD_2O$ formaldehyde | Sigma-Aldrich | #596388 |
| $NaBH_3CN$ sodium cyanoborohydride | Sigma-Aldrich | #296945 |
| Undecanal | Alfa Aesar | #A16101 |
| Chromabond HR-X 20 mg spin-columns | Macherey-Nagel | #730525 |
| Chromabond HR-X Multi 96 50 mg | Macherey-Nagel | #738530 |
| C1r | Complement Technology | #A102 |
| C1s | Complement Technology | #A104 |
| Trypsin | Serva Heidelberg, Germany | #37286.03 |
| Wieslab Assay Alternative Pathway | SVAR Life Science, Sweden | #COMPLAP330RUO |
| Wieslab Assay Classical Pathway | SVAR Life Science, Sweden | #COMPLCP310RUO |
| Wieslab Assay MBL Pathway | SVAR Life Science, Sweden | #COMPLMP320RUO |
| Dynabeads His-tag Isolation and Pull-Down | Invitrogen | #10103D |
| MagReSyn Ti-IMAC HP | ReSyn Biosciences | #MR-THP010 |
| IL6 | Invivogen | #7270-IL/CF |
| MitoTracker Deep Red | Thermo/Invitrogen | #M22426 |
| Hoechst 33342 | Thermo/Invitrogen | #H3570 |
| Concanavalin-Alexa Fluor 488 conjugate | Thermo/Invitrogen | #C11252 |
| SYTO 14 Green Fluorescence Nuclei Acid Stain | Thermo/Invitrogen | #S7576 |
| Phalloidin-Alexa Fluor 568 conjugate | Thermo/Invitrogen | #A12380 |
| Wheat-Germ Agglutinin-Alexa Fluor 555 conjugate | Thermo/Invitrogen | #W32464 |
| **Software** | | |
| FragPipe | (Yu et al, 2023) | V20.0 with built-in DiaNN v1.8.1beta2 or V21.1 with built-in DiaNN v1.8.1beta8 for $^{13}C_6$-lysine dataset analysis |
| Spectronaut | Biognosys | V19.0.240606.62635 |

| Reagent/Resource | Reference or Source | Identifier or Catalog Number |
|---|---|---|
| MANTI | (Demir et al, 2021) | V5.5 |
| R | (R Core Team, 2022) | V4.3.1 |
| circos | (Krzywinski et al, 2009) | V0.69-9 |
| MOFA+ | (Argelaguet et al, 2020) | V1.10.0 |
| Zen | Zeiss | V3.0 |
| CellProfiler | (Lamprecht et al, 2007) | V2.1.1 |
| cytominer | (Becker et al, 2017) | V0.1.0 |
| gplots | (Warnes et al, 2024) | V3.03 |
| corrplot | (Wei and Simko, 2021) | V0.84 |
| **Other** | | |
| Opentrons OT-2 | Opentrons | |
| Thermo PepMap 100 C$_{18}$ (#164535, 2 cm) pre-column | Thermo Fisher | #164535 |
| Aurora Ultimate 25 cm column (#AUR3-25075C18) | ionopticks | #AUR3-25075C18 |
| Ultimate3000 RSLC | Thermo Fisher | |
| Exploris480 mass spectrometer with FAIMS Pro | Thermo Fisher | |
| AU-CD beamline at the ASTRID2 synchrotron radiation facility | Aarhus University, ISA, Dept. of Physics and Astronomy | |
| Celldiscoverer 7 microscope with AxioCam 702 CMOS 12-bit camera | Zeiss | |

## Methods & protocols

### N-terminomic High-efficiency Undecanal-based N-Termini EnRichment method development

We optimized our well-known and widely adapted High-efficiency Undecanal-based N-Termini EnRichment (HUNTER, Weng et al, 2019) method to make N-termini enrichment fast and feasible for large cohorts. Human EDTA plasma samples were diluted (1:10) and lysed in 4% SDS, 0.1 M HEPES pH 7.4, 2.5 mM EDTA, supplemented with Roche cOmplete protease inhibitors for 5 min at 95 °C and further processed in 96-well plates. Following lysis, carbamidomethylation was performed with 5 mM tris(2-carboxyethyl)phosphine (TCEP) and 20 mM chloroacetamide (CAA) for 5 min at 95 °C, quenched with a further 10 mM TCEP for 5 min at 95 °C, and proteins were cleaned up with SP3 beads (Fig. 2A). Proteins were purified using SP3 paramagnetic beads (hydrophilic:hydrophobic, 1:1 mixture) in final 80% ethanol (EtOH) for 10 min and washed two times with 200 µL 90% acetonitrile (ACN). After resuspension in 100 µL 4%SDS/0.1 M HEPES pH 7.4/2.5 mM

EDTA, the proteins were dimethyl-labeled at lysine residues using a single label approach, coined simplex HUNTER (SHUNTER). Proteins were isotopically labeled with $^{13}CD_2O$ formaldehyde on protein-level to label the free N-terminal amine groups generated by in vivo proteolytic activity. Labeling reactions were performed in two rounds for 5 min each at 95 °C with 30 mM $^{13}CD_2O$ and 15 mM sodium cyanoborohydride ($NaBH_3CN$) and quenched with 100 mM Tris-HCl pH 7.4 for 5 min at 95 °C. Subsequently, proteins were purified with a second round of SP3 bead cleanup (2x 200 µL ACN 90% + 1x 100 µL ACN 90% washes) and digested to peptides o/N at 37 °C with a trypsin:proteome ratio of 1:100 (w/w) in 50 mM HEPES pH 7.4/2.5 mM $CaCl_2$.

After taking a proteome aliquot (pre-SHUNTER sample, typically 10% of the whole sample), the remaining digested peptides were labeled with the aldehyde undecanal at a 20:1 excess (w/w) in final 40% ethanol with 30 mM $NaBH_3CN$ for 5 min at 95 °C. This step labels the free, not formaldehyde protected amine-groups of the trypsin protease-digestion generated peptides with the aldehyde undecanal. Additional 30 mM $NaBH_3CN$ was supplemented for another 5 min at 95 °C and excess undecanal was quenched with final 100 mM Tris-HCl pH 7.4 for 5 min at 95 °C. Undecanal-labeled, trypsin-generated peptide N-termini feature an increased hydrophobicity, are depleted by centrifugation in HR-X 96-well plates (50 mg, Macherey Nagel) or HR-X spin columns (20 mg, Macherey Nagel), and separated from the flow-through, the $^{13}CD_2O$-labeled protein N-termini, which were generated by proteolytic activity in vivo. The purified protein N-termini are vacuum dried and purified by SP3 bead cleanup on peptide level for enhanced sensitivity (Hughes et al, 2014; Höhne et al, 2018). The final N-terminal peptides were re-suspended in 24 µL 5% DMSO, separated from the beads and acidified by final 1% formic acid (FA) before the purified peptide concentration was determined spectroscopically using a NanoDrop One (Thermo Fisher) based measurement at $A_{280}$.

### Eculizumab treatment of patients and plasma collection

As part of the ECUSTEC trial (EudraCT Number: 2016-000997-39, Ives et al, 2024), UK pediatric patients ($n = 3/4$) presenting with Shiga Toxin producing Escherichia Coli (STEC) Hemolytic Uremic Syndrome (HUS, STEC HUS) were given Eculizumab at Day 1 and Day 8 following hospital admission. Blood samples, urine samples and clinical data were collected up to 1 month post admission. Prior to inclusion, patients provided written informed consent. The trial was conducted in accordance with the principles of the Helsinki Declaration and was approved by North East – Newcastle and North Tyneside 1 Research Ethics Committee. The EDTA-plasma samples had never been thawed before.

### Human Lupus cohort

Patients diagnosed with SLE and under the care of the outpatient clinic at the Department of Rheumatology, Aarhus University Hospital (AUH), were consecutively enrolled between 2015 and 2017. Detailed inclusion and exclusion criteria and comprehensive clinical data have been previously outlined (Troldborg et al, 2018b), but a brief summary can be found in Dataset EV6. Prior to inclusion, patients provided written informed consent. The study was conducted in accordance with the principles of the Helsinki Declaration. It was approved by both the Danish Data Protection Agency and the Central Denmark Region Committees on Health

Research Ethics (approval number: #1-10-72-214-13). The bio-banked EDTA-plasma samples ($n = 143$ for this study as well as $n = 23$ controls) had been stored at $-80$ °C and never been thawed.

### N-termini sample preparation from cross-sectional SLE cohort

We processed human EDTA-plasma samples in a semi-automated manner on an Opentrons OT-2 pipetting robot (Opentrons, NYC, USA) from 143 SLE patients of the AUH SLE cohort (Troldborg et al, 2018b) and age-matched healthy control subjects ($n = 23$). In brief, previously unthawed EDTA plasma was diluted 1:8 with 4% SDS/0.1 M HEPES pH 7.4/5 mM EDTA, supplemented with Roche cOmplete protease inhibitor cocktail (1x), and was denatured for 5 min at 95 °C. Subsequently, the protein concentration was determined by the Pierce bicinchoninic acid (BCA) protein assay (Smith et al, 1985) in a 1:10 dilution and the same amount of 250 µg for all of the human EDTA-plasma was processed as indicated under N-terminomic High-efficiency Undecanal-based N-Termini EnRichment method development, with the following adaptations: cysteines were carbamidomethylated with 10 mM TCEP and 50 mM CAA for 5 min at 95 °C. Excess CAA was quenched with a second round of 10 mM TCEP for another 5 min at 95 °C. Proteins were purified using SP3 beads and dimethyl-labeled at lysine residues with $^{13}CD_2O$ before a second round of SP3 bead purification was performed. After air-drying the beads for a brief amount of time (1 min), the beads were re-suspended in 60 µL 50 mM HEPES pH 7.4/2.5 mM $CaCl_2$, supplemented with 1:100 trypsin (Serva #37286.03, Heidelberg, Germany) and digested o/N at 37 °C at 1000 rpm.

Following digestion o/N, a bulk proteome sample was obtained by taking a 10% preSHUNTER aliquot, purified by $C_{18}$ Stage-Tip cleanup. The remaining 90% was used for hydrophobic tagging of the newly generated tryptic peptide N-termini with undecanal (20:1 undecanal-to-protein ratio). To purify the original dimethyl-labeling blocked protein-N-termini from the aldehyde-labeled tryptic peptide N-termini, a C18 depletion was performed using HR-X 96 well plates (50 mg, Macherey-Nagel) in 40% EtOH. The combined flow-through is evaporated in a vacuum concentrator, purified using SP3 beads and the final N-terminal peptides' concentration determined using a NanoDrop One (Thermo Fisher) measurement at $A_{280}$.

### Longitudinal analysis of lupus nephritis patient plasma: active disease vs. remission

Patients with lupus nephritis ($n = 6$, 4 females) from the Hamburg Glomerulonephritis registry (University Hospital Hamburg-Eppendorf, Hamburg, Germany) were used for monitoring the active vs. remission states of lupus nephritis according to KDIGO. Prior to inclusion, patients provided written informed consent. The study was conducted in accordance with the principles of the Helsinki Declaration and was approved by the Hamburg Committee on Health Research Ethics (approval number: PV4806). Only patients with biopsy proven class IV OR III/IV OR III/IV with or without class V were chosen. Initial serum samples were taken at a time (mean age $38.5 \pm 9.0$ (SD)) when patients had active disease. Subsequent samples were taken when patients had reached complete renal response (CRR) according to KDIGO: reduction in proteinuria <0.5 g/g measured by urinary Protein Creatinine Ratio (uPCR) and stabilization or improvement in kidney function (±10–15% of baseline). Clinical data and samples were gathered at

the same time for analysis of proteolysis. Proteolysis analysis was performed according to the SHUNTER workflow for the SLE cohort. The regulation of proteolytic processing was quantified as the alteration in the N-termini abundance between active/remission states in the same patient and averaged over the patient group.

### SILAC mouse serum for labeling

Male Bl6/N mice (12-week-old) were fed a custom diet (Silantes, Germany) containing more than 99% $^{13}C_6$ lysine. The protocol for labeling was described previously (Rinschen et al, 2022; Warscheid, 2014): after one and two weeks (protocol 2, $n = 3$ biological replicates), mouse serum was collected and snap-frozen and stored at $-80\,°C$. Mouse serum was handled and measured according to the SHUNTER workflow.

### Proteases for in vitro assay

C1r and C1s were plasma-derived, commercial products: C1r (Complement Technology, #A102), C1s (Complement Technology, #A104). The production of the active MASP-1 and the inactive MASP-1 we use in the present report are already described in detail (Pihl et al, 2021); they are denominated MASP-1 EK and MASP-1 EK S646A, respectively, in that publication. The production of the active MASP-3 and the inactive MASP-3 we use in the present report are described in detail elsewhere (Pihl et al, 2017); they are denominated MASP-3 and MASP-3 S664A, respectively, in that publication.

### In vitro assays for protease profiling in the complement system

To obtain information on the targets for selected members of the complement system, in vitro assays were performed with human EDTA-plasma from healthy individuals ($n = 4$, 2 females and 2 males) in a 1:85 ratio with recombinant proteases (5 μg protease on 425 μg plasma proteome): C1r and C1s representing the classical, MASP-1 the lectin and MASP-3 representing the alternative pathway including corresponding inactive controls (Megyeri et al, 2013; Degn et al, 2014) comprised of active site mutants for MASP-1 and MASP-3 (S646A). Human plasma was diluted in 50 mM HEPES pH 7.4/2.5 mM $CaCl_2$ to ensure the same protein concentrations in all individual samples and supply subsequent protease activity with $Ca^{2+}$, immediately heat-inactivated at $56\,°C$ for 30 min to inhibit intrinsic proteolytic activity (Fante et al, 2021), and subjected to treatment with respective proteases for 2 h at $37\,°C$. Accordingly, active and inactive, recombinant MASP-1 and MASP-3 proteases were incubated with purified human C3 ($n = 3$ technical replicates) in a low protease:protein (w/w) ratio of 1:50 and a high protease:protein ratio of 1:10 in 50 mM HEPES pH 7.4/2.5 mM $CaCl_2$ ($37\,°C$ for 2 h) to reveal direct cleavages by MASP-1/3 without any contribution of other human plasma components. After the recombinant protease incubation, the samples were immediately denatured for 5 min at $95\,°C$ to diminish all proteolytic activity and the SHUNTER workflow was applied. The identified N-termini from the active proteases were compared to catalytically inactive variants, or control, non-treated samples and were used to identify specific protein cleavages generated by the proteases (cut-off $\log_2 FC > 1$ & limma moderated t-test $p$-value < 0.05 or novel to protease treatment). Bioinformatically reconstructed N-terminal cleavage windows of P6'-P6 encompassing the identified N-termini against all identified N-terminal cleavage windows in the corresponding experiments were used to generate iceLogos (Colaert et al, 2009) depicting the cleavage specificity.

Additionally, the effect of C3-LHF1 on the MASP-1 activity was investigated in a triplex labeled in vitro digestion assay with human plasma ($n = 4$ individuals). In brief, 200 μg of human EDTA-plasma was heat-inactivated (30 min, $56\,°C$) and incubated with inactive MASP-1/-3, active MASP-1/-3 or active MASP-1/-3 supplemented with 20 μM C3-LHF1 for 2 h at $37\,°C$ in a 1:85 protease:protein ratio in 50 mM HEPES pH 7.4/2.5 mM $CaCl_2$. The N-termini were enriched according to the SHUNTER protocol and for each individual plasma sample, the conditions were labeled differentially with isotopically distinct formaldehyde and subsequently pooled: inactive MASP-1 represented the light channel, (+28.0313 Da), active MASP-1 supplemented with 20 μM C3-LHF1 represented the medium channel (+32.0564 Da), and active MASP-1 represented the heavy channel (+36.0756 Da).

### SEC fractionation of patient plasma

For a selected SLE patient plasma sample from the SLE cohort (severe SLEDAI disease score of 10 and lupus nephritis), a Superdex 200 (GE Healthcare Life Sciences Inc.) size-exclusion chromatographic (SEC) separation was applied to yield fractionated proteins and protein complexes. The column's void and bed volumes are 7.5 ml and 24 ml, respectively. In brief, 100 μL EDTA plasma was mixed 1:1 with 20 mM HEPES pH 7.5, 140 mM NaCl, 5 mM EDTA, 0.1 mM Pefabloc in a 200 μL loop, and SEC fractions were collected in 250 μL each. Subsequently, four sequential SEC fractions ($4 \times 250$ μL $= 1$ mL) were pooled, and protein concentrations were assayed by the BCA (Smith et al, 1985) assay. Only fractions after the elution volume of 6.5 mL and before 18.5 mL contained detectable protein, and correspondingly, a total of 16 fraction pools containing detectable amounts of protein were used for SHUNTER N-termini analysis. The SEC fraction pools were processed and measured according to the procedure described for the SLE cohort samples.

### LC/MS setup for measuring

Samples were measured on a Thermo Ultimate 3000 nanoHPLC system coupled online to an Exploris480 mass spectrometer. A total of 1 μg of purified N-terminal peptides were separated in a 60-min total runtime binary LC gradient on a two-column setup with a Thermo PepMap 100 $C_{18}$ (#164535) pre-column and an Aurora Ultimate 25 cm column (#AUR3-25075C18). For most SHUNTER samples, data-independent acquisition (DIA) was used. The DIA gradient ranged from 5–25% B (A: $H_2O$ + 0.1% FA, B: ACN + 0.1% FA) for 35 min at 280 nL/min, heated at $50\,°C$. DIA MS acquisition for plasma samples was performed with the FAIMS Pro device (Standard Resolution mode with 3.8 L/min gas flow) at two alternating compensation voltages of $-40$ and $-61$ V throughout the total runtime of 60 min with MS1 acquisition being operated in positive profile mode at a resolution of 120,000 in the scan range of 350–1350 $m/z$ and a normalized AGC target of 300% (RF Lens 40%). DIA MS2 data was recorded in positive profile mode at a resolution of 30,000 with 40 windows from 350–1350 $m/z$, 25 $m/z$ each and a normalized AGC target of 300% (RF Lens 50%). Preferred charge states were set to 2–6. Parts of the runs, especially from the method development period, were performed in data-dependent mode (DDA) on the same setup. For the analysis of the kidney organoid phosphoproteomes, proteomes, and kidney pull-

downs, LC/MS analysis was performed with DIA MS as described elsewhere (Billing et al, 2024).

## Data analysis

DDA and DIA runs were recorded and analyzed by FragPipe v20.0 and the built-in DiaNN v1.8.1beta2 (Yu et al, 2023) to validate the improved SHUNTER workflow (Fig. 2A). To determine the labeling efficiency for the dimethyl labeling and the pullout efficiency for the undecanal tagging of the tryptic peptides, searches were performed (a) with trypsin specificity and the variable dimethylation on lysines ($+34.0631$ Da for $^{13}CD_2O/NaBH_3CN$) for the preSHUNTER samples or (b) with ArgC specificity and variable acetylation ($+42.0210$), dimethylation ($+34.0631$) and pyro-Glu formation ($-17.0265$ on Q/C or $-18.0106$ on E) on N-termini, fixed dimethylation on lysines ($+34.0631$ Da), carbamidomethylation on cysteines ($+57.02146$ Da) and variable oxidations on methionine ($+15.9949$ Da).

Efficiency of dimethylation on lysines was determined by comparing the percentage of labeled lysines to all lysines in the corresponding peptides containing lysine in each replicate. The pullout efficiency was defined as the percentage of labeled/protected protein N-termini compared to all identified N-termini, which do not feature any N-terminal modification. To assess the degree of protein modifications generated by the old HUNTER or the new SHUNTER protocol, the open search strategy was applied in FragPipe (Yu et al, 2020; Geiszler et al, 2021b). In brief, due to the large search windows of $-150$ to $500$ Da, any potential modification could be detected in these searches, and identified putative modifications are analyzed with PTMShepherd (Geiszler et al, 2021a) to determine the most abundant modifications and their putative localization to distinct amino acids. The information herein can be used to monitor the degree of undesired side reactions or biases in the methods HUNTER vs. SHUNTER.

Acquired RAW files for SHUNTER runs of the SLE cohorts were analyzed with FragPipe with a fixed N-termini strategy, where only dimethylated lysine and peptide N-termini ($+34.0631$ Da for $^{13}CD_2O/NaBH_3CN$) in addition to carbamidomethylation on cysteines ($+57.02146$ Da) were required as fixed modifications. Variable modifications were limited to oxidations on methionine ($+15.9949$ Da). Quantification within FragPipe/DiaNN was performed with default settings and a FDR of 0.01 (1%); DiaNN utilizes a normalization scheme of the total precursor ion intensities with the help of the LC retention time (RT) which was applied within this study. Additionally, we made sure that the quantification of N-termini in DIA as derived from semi-specific searches in FragPipe/DiaNN is reliable (cf. Appendix Figs. S7–14 for more information).

$^{13}C_6$-lysine labeled mouse SILAC samples were analyzed in FragPipe v21.1 with the help of built-in DiaNN v1.8.2_beta_8 using the library-free plexDIA approach, which was introduced in this version of DiaNN. To quantify the $^{13}C_6$-lysine vs. unlabeled $^{12}C_6$-lysine difference, a fixed N-termini search strategy was applied as above, but with additional $^{13}C_6$-labeled lysines ($+6.0200996$) set as heavy label and $^{12}C_6$ lysine ($+0$) as light label. Subsequent data evaluation and annotation were performed by MANTI (Demir et al, 2021) v5.5, adapted to handle FragPipe/DiaNN-generated DIA N-termini data. Data analysis and visualization utilized circos v0.69-9 (Krzywinski et al, 2009) and R v4.3.1 (R Core Team, 2022).

Analysis for shotgun proteome samples (human kidney organoids, pull-down assays) was performed with Spectronaut v19.0.24060.62635 (Biognosys) with default settings (directDIA analysis, 1% FDR, no imputation) against the canonical human reference proteome (UniProt, 2022-09, 20,600 entries). Following modifications were set for the database query: fixed carbamidomethylation on cysteines ($+57.02146$ Da), variable oxidation of methionine ($+15.9949$ Da), variable acetylation of the protein N-terminus ($+42.0210$ Da), and for phospho-specific searches: phosphorylation on serine, threonine and tyrosine ($+79.9663$ Da).

## Multi-Omics-Factor Analysis (MOFA)

The quantified data was analyzed with the MOFA2 (Argelaguet et al, 2020) package v1.10.0 in R. Log10-transformed intensities for N-termini, and proteins were used as quantitative traits and the corresponding gene ID as a feature for proteins. On the other hand, N-termini were identified by a feature-tag consisting of a "gene:start_position-end_position" (e.g., C3:755-780) identifier. MOFA2 model generation was restricted to ten factors, and the convergence level for model generation was "slow" to yield the highest quality for the MOFA2 model. Putative candidate cleavages were selected by a substantial regulation in the SLE cohort versus the controls of at least 27% ($|\log_2FC| > 0.35$, cf. Fig. EV3A) and a substantial contribution to individual MOFA2 model factors (within the top 25%).

## Synthesis of C3-LHF1 recombinant fragment

The protein fragment C3-LHF1 (C3-Lupus Human Fragment 1) was produced in mammalian CHO cells and represents a 156 amino acid human C3 fragment (pos. 1514-1663) with an additional C-terminal 6x His-tag (MGWSCIILFLVATATGVH-SAEENCFIQKSDDKVTLEERLDKACEPGVDYVYKTRLVKVQLS NDFDEYIMAIEQTIKSGSDEVQVGQQRTFISPIKCREALKLEEK KHYLMWGLSSDFWGEKPNLSYIIGKDTWVEHWPEEDECQDE ENQKQCQDLGAFTESMVVFGCPNHHHHHH—additional tags for secretion and purification underlined) and was correspondingly purified. Endotoxins were removed and validated to be ≤0.1 EU/mg by the manufacturer.

## Circular dichroism for secondary structure assessment of C3-LHF1

Recombinantly expressed C3-LHF1 was dialyzed in a Slide-a-Lyzer dialysis cassette with a 3.5 kDa MWCO (molecular weight cut-off) against 20 mM sodium phosphate buffer pH 7.4 with 150 mM NaF to eliminate residual NaCl from the protein preparation. The synchrotron radiation circular dichroism (SRCD) spectrum of C3-LHF-1 was acquired at the AU-CD beamline at the ASTRID2 synchrotron radiation facility (ISA, Department of Physics and Astronomy, Aarhus University, Denmark).

The CD spectrum of the LHF-1 protein was measured at 25 °C over the wavelength range 170 to 280 nm in 1 nm steps with a dwell time of 2 s/pt, repeating to record 6 scans. The sample was at a concentration of 0.45 mg/ml in a 20 mM potassium phosphate buffer at pH 7.0, 150 mM KF. An appropriate reference baseline was measured in triplicate and subtracted from the averaged sample spectrum. Measurements were carried out using a nominally 0.01 cm pathlength quartz cell (SUPRASIL, Hellma GmbH, Germany. The actual pathlength of the cell was determined to be 0.01023 cm by an interference technique (Hoffmann et al, 2016). The differential molar extinction coefficient, $\Delta\varepsilon$, spectrum was calculated using protein concentration determined from the

absorbance at 205 nm (measured simultaneously with the CD spectrum) and the protein molar extinction coefficient at that wavelength (Anthis and Clore, 2013). The secondary structure content of the protein from the spectrum was estimated using python code implemented by the SRCD group at Aarhus University. The code utilizes the CDSSTR analysis routine, using a SP175 reference set curated from the PCDDB (Whitmore et al, 2011) and secondary structure components determined via DSSP analysis (Kabsch and Sander, 1983) of the associated PDB crystal structures.

The C3-LHF-1 spectrum recorded at 25 °C exhibits a mixed contribution of ordered α-helical and β-sheet structures resulting in a maximum at 193 nm and a single minimum at 210 nm, with $\Delta\varepsilon$ ranging between ~1.5 and $-3\,M^{-1}\,cm^{-1}$, indicating a significant contribution of unordered structures as well. Spectral fit analysis revealed $25 \pm 5\%$ β-sheet, $13 \pm 4\%$ turns and $39 \pm 6\%$ other unordered structures, which is consistent with the C345c domain of C3. The experimental data was then fitted using the CDSSTR analysis program and AU-SP175 reference set. The fit of the spectrum gave the following secondary structure results, $17 \pm 3\%$ α-helix, $25 \pm 5\%$ β-sheet, $13 \pm 4\%$ turns and $39 \pm 6\%$ other unordered structures. This compares well with the expected secondary structure from the crystal structure of the C345c domain of complement C3, which is modeled via DSSP to have ~28% α-helix, ~33% β-sheet and ~39% other structures. This is evidence that the recombinant LHF1 recombinant fragment has an overall secondary structure content similar to the C345c domain of complement C3, likely due to the same tertiary folding, validating further assays with the recombinant protein.

### Wieslab complement ELISA assays

To assess any modulation exhibited by the C3-LHF1 fragment, we used the Wieslab® Complement System Screens (SVAR Life Science, Malmö, Sweden) for the Alternative, Classical, and MBL pathways. The assay detects the deposition of complement factor C9 on the three different surfaces used in the kit. In brief, freshly thawed human serum was pre-incubated with increasing concentrations of C3-LHF1 (1, 2, 5, 10, and 20 µM) or 1x PBS as control, all diluted to the same volume with the corresponding diluent buffers from each kit for 30 min at RT. Subsequently, the assay was performed as instructed by the manufacturer, and the ELISA plate read at 405 nm. Corresponding samples with no detectable activity in each pathway were eliminated for further data analysis.

### Assay for deposition of C3 fragments via mannan-binding lectin

The commercial Svar Wieslab assay (used above) detects the deposition of C9 onto different surfaces. The effect on the mannan-binding lectin pathway of the complement system when adding compounds to human serum was further examined by measuring the impact on the deposition of C3 fragments onto mannan-coated microtiter well surfaces. Microtiter wells were coated with 1 mg mannan (M7504; Sigma-Aldrich) in 100 µl 50 mM sodium carbonate, pH 9.6. Residual protein-binding sites were blocked by incubation of TBS (10 mM Tris, pH 7.4, 145 mM NaCl, 0.05% Tween 20) containing 1 mg/ml human serum albumin (HSA) for 1 h at room temperature, followed by washing three times in TBS/Tween (TBS with 0.05% Tween 20). Then we add 100 µl normal human serum (NHS) diluted to 1% (v/v) in 10 mM HEPES, 140 mM NaCl, 2 mM CaCl₂, 1 mM MgCl₂, pH 7.5, in the presence

of C3-LHF1 in a dilution series. We also included a nanobody (hC3Nb2) that inhibits C3 fragment depositions (Jensen et al, 2018) as a control. After incubation of the mixtures for 30 min on ice, the samples were added to the wells, and the plate was subsequently incubated for 1.5 h at 37 °C in a humid chamber and then washed three times in TBS/Tween. Next, 100 µl 0.75 mg/ml biotinylated anti-hC3d (catalog no. A0063; Dako) in TBS/Tween was added to the wells, followed by a 2-h incubation period at room temperature. The wells were subsequently washed three times in TBS/Tween, then incubated with 1 mg/ml streptavidin-europium (PerkinElmer) in TBS/Tween, 25 µM EDTA, for 1 h at room temperature. Subsequently, the plate was washed three times in TBS/Tween, followed by incubation with enhancement buffer (Ampliqon) for 2 min. The signal of the europium in the wells was subsequently read by time-resolved fluorometry on a plate reader (VICTOR, PerkinElmer) as counts per second.

### Assay for alternative pathway (AP) by hemolysis

We analyzed the effect of the peptides on alternative pathway mediated hemolysis of rabbit erythrocytes. For this, 11% normal human serum, diluted in 20 mM HEPES pH 7.4, 150 mM NaCl, 5 mM MgCl₂, 10 mM EGTA with 0.1% gelatin (hemolysis buffer), was mixed with dilutions of C3-LHF1. The EGTA/Mg buffer allows the alternative pathway to be active, whereas lack of calcium ions inhibited the lectin pathway and the classical pathway. A positive control with a nanobody that inhibits activation of C3 was included as a control. Then 20 µl of these mixtures and 10 µL 6% v/v rabbit erythrocytes in hemolysis buffer were added to V-bottom microtiter wells (Thermo, 249662). The plate was incubated at 37 °C for 2 h, and then complement activity was stopped by adding 60 µL of cold 145 mM NaCl, 5 mM EDTA. The non-lysed erythrocytes were pelleted by centrifugation at $200 \times g$ for 20 min and 70 µL of the supernatant was transferred to a 96-well plate. The absorbance was measured at 405 nm on a plate reader.

### Pull-down assays from human plasma with C3-LHF1

Human EDTA-plasma ($n = 4$ healthy controls $+$ $n = 4$ SLE patients, each 2x male and 2x female) was diluted with 1x PBS, 0.02% Tween-20, 1x cOmplete protease inhibitors without EDTA (Roche) to the same protein concentrations and pre-cleared on NTA-Co²⁺-based magnetic beads (Invitrogen) for 30 min with gentle rotation (15 rpm) at RT. Meanwhile, the beads for the pull-down were prepared by immobilizing (i) recombinant His-tagged C3-LHF1, (ii) His-tagged nanobody against C3 (hC3Nb2), and (iii) a blank negative control (NC) on NTA-Co²⁺-based magnetic beads for 10 min at RT. After the pre-clearing, each unbound human plasma was equally split into three assays for C3-LHF1, hC3Nb2 and NC and a small aliquot was kept for proteomic analysis of each pre-cleared lysate. The pull-down was performed for 1 h with gentle rotation (15 rpm) and beads subsequently washed 4x with 1x PBS, 0.02% Tween-20, 600 mM NaCl, 1x cOmplete protease inhibitors without EDTA. Elution was performed with 1x Lämmli buffer $+$ 0.5 M imidazole for 30 min at 37 °C and the eluate was purified with SP3 beads for proteomic sample processing (o/N digestion with trypsin at 37 °C).

### Pull-down assays from human kidney with C3-LHF1

Healthy human kidney tissue was homogenized in 1x RIPA buffer supplemented with Roche cOmplete protease inhibitors and lysed

for 30 min on ice before centrifugation at $15,000 \times g$ (4 °C, 15 min). The lysate was split into $n = 3$ technical replicates and pre-cleared on NTA-Co$^{2+}$-based magnetic beads (Invitrogen) for 30 min with gentle rotation (15 rpm) at RT as for the human plasma pull-down assays. The pre-cleared, non-bound lysates were equally split among bead control, C3-LHF1 and C3-LHF1$^{DN}$ pull-downs and incubated for 30 min with gentle rotation (15 rpm) at RT. After excessive washing with 50 mM KP$_i$ pH 8.0/300 mM NaCl/0.01% Tween-20, beads were resuspended in 50 µL 50 mM HEPES pH 7.4/2.5 mM CaCl$_2$ and pull-downs digested o/N on bead with trypsin at 37 °C. Digested peptides were cleaned up for MS analysis with SP3 beads.

### Thermal proteome profiling - proteome integral solubility alteration assay for C3-LHF1

Thermal proteome profiling - proteome integral solubility assays (TPP-PISA, Batth et al, 2024) were performed on human kidney tissue ($n = 3$ technical replicates) lysed in 50 mM HEPES pH 7.4/100 mM NaCl/2.5 mM EDTA pH 8.0/Roche cOmplete protease inhibitors/20 µg/mL DNAse I. Treatments with 1 or 10 µM C3-LHF1 or 1xPBS control were carried out at 37 °C for 1 h and subsequent thermal denaturation was performed at three different temperatures (53/56/59 °C) for 5 min before the samples were combined. Combined proteins were solubilized with 0.8% NP-40/50 mM HEPES pH 7.4/100 mM NaCl/2.5 mM EDTA pH 8.0/Roche cOmplete protease inhibitors (30 min at 4 °C) and the soluble proteins were separated by centrifugation at $18,000 \times g$ for 30 min (4 °C). An aliquot of 50 µg from the soluble proteome supernatant was lysed in final 2% SDS/50 mM HEPES pH 7.4/2.5 mM EDTA for 5 min at 95 °C, carbamidomethylated with 5 mM TCEP/20 mM CAA for 5 min at 95 °C, purified with paramagnetic SP3 beads and digested o/N at 37 °C with trypsin (protease:proteome ratio 1:50). Subsequently, the peptides were purified using C$_{18}$ StageTips before being analyzed via HPLC/MS.

### IL6 reporter assays

HEK-Blue IL-6 reporter line cells (Invivogen #hkb-hil6) reconstituting the IL6Rα/IL6ST/STAT3 axis and exhibiting a secreted embryonic alkaline phosphatase (SEAP) response to activation were cultivated according to the manufacturer's instructions. In Brief, cells were seeded in a volume of 100 µL DMEM media supplemented with glutamine and high glucose onto a 96-well plate. Incubations were performed with 100 µL of fresh media, supplemented with recombinant human IL6 (R&D Systems #7270-IL/CF) as a positive control, reconstituted in 1x PBS, or C3-LHF1 o/N at 37 °C, 5% CO$_2$. Subsequently, 180 µL of QUANTI-Blue solution were added to 20 µL of the cells, the reaction incubated for 1 h at 37 °C and the colorimetric SEAP product measured at 639 nm using a spectrophotometer. Inhibition of the IL6Rα response was performed by pre-incubating some wells with Tocilizumab at 2 µg/mL or vehicle for 3 h at 37 °C before performing the subsequent treatment. Inhibiting the C3-LHF1 activity was achieved by pre-incubation of C3-LHF1, hC3Nb3 or a combination of C3-LHF1 with the hC3Nb3 nanobody in a 1:1 w/w ratio at 37 °C for 1 h before treatments were started on the cells.

### Structural mapping

Mutations were mapped onto the structures of native C3 from the Protein Data Bank (PDB, Berman et al, 2000) entry 6ru5, C3b

(entry 6ehg), C4 (entry 4jpm), C4b (6ysg) and C8a,b,c (entry 3ojy, Lovelace et al, 2011) using pymol. In C4 and C4b, the alpha- and gamma-chains were colored similarly for clarity.

### Granulocyte CD62L shedding assay

Activation of granulocytes (CD62L shedding assay) was conducted using samples of four milliliters of sodium heparin-stabilized whole blood obtained from healthy blood donors ($n = 5$ individuals) at the blood bank of Aarhus University Hospital, Denmark. Experiments commenced within 1 h of blood collection. The samples were gently mixed, and fifty microliters were transferred into five-milliliter polystyrene tubes. Subsequently, each tube was treated with C3-LFH1 (1.2 mg/mL = 66 µM), POD2 (irrelevant control peptide, 1.7 mg/mL = 1 mM), lipopolysaccharide (LPS, 0.19 mg/mL, sourced from Escherichia coli 0111:B4, 'tlrl-3pelps', Invivogen, San Diego, CA, USA), or sterile phosphate-buffered saline (PBS, pH 7.4). All compounds were dissolved in sterile PBS, with final concentrations as specified in the text. After gentle mixing, the tubes were incubated in a water bath at 37 °C for 1 h.

For each donor, all tubes, except one treated with PBS, received 5 µL of allophycocyanin-labeled mouse monoclonal IgG anti-human CD62L (Clone: DREG-56, Catalog No: 559772, BD Pharmingen, Franklin Lakes, NJ, USA). After gentle mixing, the tubes were incubated in the dark for 15 min at ambient temperature. Subsequently, each tube was treated with 1 mL of erythrocyte lysis buffer (composed of 155 mM ammonium chloride, 10 mM potassium hydrogen carbonate, and 0.1 mM EDTA, pH 7.3), gently mixed, and then incubated in the dark for an additional 15 min at ambient temperature.

The presence of surface CD62L on granulocytes was then assessed using flow cytometry. Cell suspensions were analyzed following incubation using a NovoCyte 3000 Flow Cytometer (ACEA Biosciences, CA, USA). Allophycocyanin was excited at 640 nm, and emissions were detected using a 675/30 nm bandpass filter. Analysis was conducted on 100 microliters from each tube. Granulocytes were gated based on forward- and side-scatter signals. Allophycocyanin fluorescence was analyzed as height signals.

### Cell painting assay

Human bone osteosarcoma cells, U-2OS (ATCC HTB-96) were cultured in McCoy's 5A (Sigma cat. no. M9309) supplemented with 10% fetal bovine serum (FBS, Gibco cat. no. A3160802) and 1% penicillin/streptomycin. Cells were cultured at 37 °C in a humidified atmosphere (5% CO$_2$) and passaged when 70–90% confluence was reached. To passage cells, they were washed in Dulbecco's phosphate buffered saline (PBS, $2 \times 5$ mL, Sigma, cat. no. D8537) and detached from the culture flask by trypsin-EDTA (Sigma, cat. no. T4049) and one portion of cells are reseeded in fresh full growth medium in a T75 flask (Thermo Scientific cat. no. 130190). The cell painting protocol is adapted to a 96-well plate format (Bray et al, 2016).

Cells (4000 cells/well) were seeded into the inner 60 wells of a 96-well plate with optical bottom (Corning Cat# 3603) in complete medium (75 µL) and incubated (37 °C, 5% CO$_2$, humid) for 24 h. C3-LHF1 is dissolved in water. To ensure comparability, stocks of C3-LHF1 were first diluted into media containing DMSO (50 µL C3-LHF1 stock + 75 µL 3.33% DMSO in media) to make 4X solutions with 2% DMSO (final DMSO = 0.5%). All other compounds were dissolved in DMSO and diluted 50 times in media to make 4X solutions with 2% DMSO. Compounds or

DMSO (for negative control) were dosed in the designated culture plates in quadruplicates, distributed over 4 plates, in 25 μL medium with a normalized DMSO concentration (0.5%). Twelve DMSO control wells were included on each plate for normalization. After 24 h, 75 μL medium was removed and replaced with 75 μL complete medium containing 500 nM MitoTracker Deep Red (final C = 325 nM), and plates were incubated in the dark for 30 min. Wells were then aspirated, 75 μL medium was added, before adding 25 μL 16% paraformaldehyde (Electron Microscopy Sciences 15710-S, final PFA = 4%), and plates were incubated in the dark for 20 min. Plates were washed once with 1X HBSS (Invitrogen Cat#: 14065-056) and 75 μL 0.1% (vol/vol) Triton X-100 (BDH Cat#: 306324N) in 1X HBSS was added, and incubated for 15 min. in the dark. Plates were washed twice with 1X HBSS before addition of 75 μL multiplex staining solution (Hoechst 33342: 5 μg/mL; Concanavalin-Alexa Fluor 488 conjugate: 35 μg/mL; SYTO 14 Green Fluorescence Nuclei Acid Stain: 3 μM; Phalloidin-Alexa Fluor 568 conjugate: 5 μL/mL; Wheat-Germ agglutinin-Alexa Fluor 555 conjugate: 1.5 μg/mL) in HBSS containing 1% BSA (Sigma-Aldrich Cat# A9647) and incubation for 30 min in the dark. Plates were washed three times with 1X HBSS with no final aspiration and imaged immediately in a Zeiss Celldiscoverer 7 automated microscope.

Nine images are acquired in each well with $2 \times 2$ binning using the AxioCam 702 CMOS 12-bit camera with 4x analog gain in Zen 3.0 software for Celldiscoverer 7 using the following imaging settings:

| Channel | Dyes | Excitation LED | Beamsplitter | Emission filter |
|---|---|---|---|---|
| DNA | Hoechst 33342 | 385 nm | RTBS 405 + 493 + 610 | TBP 425/30 + 524/50 + 688/154 |
| ER | Concanavalin-AF488 | 470 nm | RTBS 405 + 493 + 610 | TBP425/30 + 524/50 + 688/145 |
| RNA | SYTO 14 green fluorescent nucleic acid stain | 511 nm | RTBS 450 + 538 + 610 | TBP 467/24 + 555/25 + 687/145 |
| AGP | Phalloidin-AF568 + Wheat-germ agglutinin-AF555 | 567 nm | RQBS 405 + 493 + 575 + 653 | QBP 425/30 + 514/30 + 592/25 + 709/100 |
| Mito | MitoTracker Deep Red | 625 nm | RQBS 405 + 493 + 575 + 653 | QBP 425/30 + 514/30 + 592/25 + 709/100 |

The workflow (Svenningsen and Poulsen, 2019) was followed to generate the bioactivity profiles. In short, CellProfiler (Lamprecht et al, 2007) 2.1.1 was used to correct images for uneven illumination, followed by image segmentation and extraction of 1476 features across nuclei, cytoplasm, and the whole cell on a per-cell basis. Features were then averaged to per-well profiles, after which the data was normalized on a per-plate basis followed by per-treatment aggregation, which affords the final profiles using the cytominer 0.1.0 package (Becker et al, 2017) in R 3.6.0 (R Core Team, 2022).

The heatmap of morphological profiles is visualized with heatmap.2 in the gplots 3.0.3 package (Warnes et al, 2024). The Pearson correlation matrix is calculated using the stats package in R 3.6.0 and visualized using the corrplot 0.84 package (Wei and Simko, 2021).

Hierarchical clustering of the correlation matrix uses the stats package and Pearson correlation coefficients as distance metric and average linkage method.

The activity score is calculated as the intra-replicate correlation (Cuccarese et al, 2020), which is the Pearson correlation between technical replicates. Significant activity is determined as intra-replicate correlation >0.6, as determined by evaluation of in-house generated profiles in consolidation with the mp-value and Mahalanobis distance (Hutz et al, 2013).

The CellProfiler pipelines and R-scripts for data-processing and visualization have previously been published (Lin et al, 2021). Raw data and images can be obtained upon request.

### Human kidney organoids, C3-LHF1 treatment

Human kidney organoids were generated as previously described (Kumar et al, 2019; Lassé et al, 2023). Kidney organoids were derived from the human female induced pluripotent stem cell (iPSC) line UKEi001-A (cellosaurus ID number: CVCL_A8PR) using a modified Takasato protocol. All experimental procedures adhered to the World Medical Association's Declaration of Helsinki ethics guidelines. Human dermal fibroblasts were obtained via skin biopsy from control patients enrolled in the IndivuHeart Study, approved by the Board of Physicians Hamburg's ethical committee (Ethik-Kommission der Ärztekammer Hamburg, PV4798/28.10.2014). Informed consent was obtained from all participants without financial compensation. To generate organoids, hiPSCs were dissociated into single cells using Accutase (Gibco), seeded onto Matrigel-coated (Corning) 6-well plates (Nunc) at a density of 12,000 cells/cm² in E8 media (Thermo Fisher Scientific) with Y-27632 (10 μM, Biorbyt), and incubated overnight at 37 °C and 5% CO$_2$. This hiPSC monolayer was cultured in E6 media (Thermo Fisher Scientific) supplemented with 7 μM CHIR99021 (Sigma) from day 1 to day 4, followed by 200 ng/ml FGF9 (Peprotech), 1 μg/ml heparin (Stemcell Technologies), and 1 μM CHIR99021 from days 5 to 7. To form organoids, cells were then dissociated using Trypsin (Gibco), washed with E6 media, and centrifuged at $200 \times g$. The cell pellet was resuspended in Stage 1 media [E6 media containing 200 ng/mL FGF9, 1 μg/mL heparin, 1 μM CHIR99021, 0.1% PVA (Sigma), 0.1% MC (Sigma), 10 μM Y-27632] and transferred to 6-well plates pre-treated with Pluronic-F12 (Sigma) for low adhesion conditions (day 7 + 0). Cell aggregates spontaneously formed after rotating the culture dishes on an orbital shaker (Thermo Fisher Scientific) at 70 rpm, incubated at 37 °C and 5% CO$_2$ for 24 h. The medium was switched to Stage 2 [E6 media containing 200 ng/ml FGF9, 1 μg/ml heparin, 1 μM CHIR99021, 0.1% PVA, 0.1% MC] for another 4 days (7 + 1 to 7 + 4). From day 7 + 5 onwards, organoids were cultured in Stage 3 media [E6 media containing 0.1% PVA, 0.1% MC] until the end of the experiment. Organoids were stimulated with TNFα, and cell pellets and media were collected and analyzed as detailed above. Experiments were

done on three independent differentiations, and at the organoid age of day 23–25.

### Phosphoproteomics and proteomics analysis of C3-LHF1 signaling in organoids

Human kidney organoids were generated as previously described (Billing et al, 2024) and treated on D23 with TNFα (50 ng/mL) or C3-LHF1 (18 μM) or combinations of both for a total duration of 48 h. Supernatants were taken after 24 h and 48 h and the final organoid cell pellets (n = 4 technical replicates) were processed for proteomics and phosphoproteomics. Lysis was performed with 300 μL 4% SDS/0.1 M HEPES pH 7.4/5 mM EDTA (supplemented with Roche cOmplete protease inhibitors) for 5 min at 95 °C and the protein concentration determined with the BCA assay (Thermo Scientific). Nucleic acids in the samples were broken down with 50U benzonase/sample for 30 min at 37 °C and cysteines carbamidomethylated with 5 mM TCEP and 20 mM CAA for 5 min at 95 °C. Subsequently, proteins were purified with SP3 beads for o/N digestion with trypsin (Serva) at 37 °C in a 1:50 protease:proteome ratio. For phosphoproteomics, >1 mg of digested peptides were purified by Oasis HLB $C_{18}$ cartridges (Waters), dried down and resuspended in 200 μL of loading buffer (80% ACN, 1 M glycolic acid, 5% TFA). Phosphopeptides were bound to MagReSyn Ti-IMAC beads (ReSyn Bio) for 20 min at RT and washed with two subsequent washes of loading buffer, 80% ACN + 1% TFA, and 10% ACN + 0.2% TFA before being eluted with 1% ammonia. Enriched phosphopeptides and proteome samples were purified with $C_{18}$ StageTips and measured as described elsewhere (Lassé et al, 2023).

### Shiny application

The Shiny app was developed using R version 4.3.2 and Shiny version 1.8.0 and deployed on shinyapps.io. The heatmap visualization was generated with ComplexHeatmap version 2.18.0. In vitro values were filtered based on $\log_2 FC > 1$ and p-value < 0.05, with non-qualifying values replaced by NA. The app's code is available on GitHub at adhutz/fatihs-app. The website can be accessed at: https://ahutz.shinyapps.io/proteolySee/.

### Statistics

Statistical tests were performed as indicated in the respective figure legends without predefined criteria. All replicates are biological replicates except stated otherwise. No samples were excluded except for attrition or lack to fulfill predefined criteria (i.e., KDIGO remission criteria).

### Blinding and randomization

Mass spectrometry analyses were performed in a randomized order. Experimentalists were blinded to the sample identity when possible.

## Data availability

Proteome data for the $^{13}C_6$-lysine labeled mouse serum N-terminome is available under the PRIDE accession PXD051273 (https://www.ebi.ac.uk/pride/archive/projects/PXD051273).
Extended data regarding the human cohorts is present in the supplemental data. The corresponding authors cannot publish individual patient resolved spectral data as raw files due to the General Data Protection Regulation (GDPR) but will grant access to the human raw data upon reasonable request, following a data

transfer agreement. The Opentrons OT-2 protocols are available under https://figshare.com/s/b2fcca1a34918ed46c3c.

The source data of this paper are collected in the following database record: biostudies:S-SCDT-10_1038-S44318-025-00598-8.

## Peer review information

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

## Acknowledgements

The authors thank Peter Zipfel, Stefan Rose-John, Hans-Willi Mittrücker and Thomas Vorup Jensen for helpful discussions, Nykola Jones for assisting in CD measurements and analysis, and the AU-CD synchrotron beamline on ASTRID2, Aarhus University, for the beam time. The authors would like to acknowledge funding from the following sources: MMR was supported by the Young Investigator Award from the Novo Nordisk Foundation, grant number NNF19OC0056043, the Carlsberg Young Investigator fellowship (Semper Ardens CF21-0596), and by the DFG (RI 2811/1-1 and RI 2811/2-1, FOR2743, SFB/TRR 422 and SFB1192-project B10). AT was supported by Lundbeck Fonden grant number R264-2017-3344, and by the Danish Rheumatism Association grant number R166-A5554. This project has received funding from the European Research Council (ERC) under the European Union's Horizon 2020 Research and Innovation Program (grant agreement no. 865738 to TBP). TBP further acknowledges financial support from the Novo Nordisk Foundation (grant NNF19OC0054782) and the Carlsberg Foundation (grant CF20-0331). ST was supported by CellPAT, a Danish National Research Foundation Center of Excellence (DNRF135). TBH was supported by the DFG (CRC1192, CRC 1648, CRC/TRR 422, CRC 1453) and by the European Research Council-ERC (CureFSGS, Project: 101141768). EH received funding through the Heisenberg-Program of the DFG (DFG 414280945) and SFB1192 Immune-Mediated Glomerular Diseases (projects B1 and C1). RM received funding by German Ministry of Research (BMBF) FungoSens grant 13GW0245B and quickTLR grant 031A113B. NK was supported by the Young Talented Cancer Researchers Award from the Danish Cancer Society, grant number R389-A23037. FB is supported by the Federal Ministry of Education and Research under funding number 01EO2106 and the DFG: BR 6668/2-1). Responsibility for the content of this publication lies with the authors. Funding for the ECUSTEC trial was from the National Institute for Health and Care Research (Efficacy and Mechanism Evaluation) to SAJ. SAJ's institution has received funding from Alexion and Novartis for SAJ's participation in Advisory Boards and giving talks at symposia.

## Author contributions

**Fatih Demir**: Conceptualization; Resources; Data curation; Software; Formal analysis; Supervision; Funding acquisition; Validation; Investigation; Visualization; Methodology; Writing—original draft; Writing—review and editing. **Elina Kovalenko**: Investigation. **Moritz Lassé**: Resources; Formal analysis; Validation; Investigation; Methodology; Writing—review and editing. **Esben B Svenningsen**: Formal analysis; Validation; Investigation; Methodology. **Jens M Bernth Jensen**: Data curation; Validation; Investigation. **Anja M Billing**: Resources; Methodology. **Kathrin Groeneveld**: Data curation; Formal analysis; Validation; Investigation; Methodology. **Arvid Hutzfeldt**: Software; Formal analysis; Visualization; Methodology. **Lars Nilges**: Investigation. **João P L Guerra**: Resources; Formal analysis; Investigation; Visualization. **Krzysztof J Pietrzak-Lichwa**: Resources. **Yifan Tan**: Investigation. **Elizabeth Colby**: Resources. **Annette G Hansen**: Resources; Investigation; Methodology. **Naziia Kurmasheva**: Investigation. **David Olagnier**: Project administration. **Dongwoo

**Choi**: Investigation; Writing—review and editing. **Mika M Richter**: Methodology. **Sandra D Laufer**: Methodology. **Fabian Braun**: Resources. **Sally A Johnson**: Resources. **Marcus Krüger**: Resources. **Tobias B Huber**: Resources. **Elion Hoxha**: Resources. **Oliver M Steinmetz**: Resources. **Ralf Mrowka**: Resources; Methodology. **Simon Melderis**: Resources. **Moin A Saleem**: Conceptualization; Resources. **Thomas B Poulsen**: Resources; Funding acquisition; Methodology. **Gregers R Andersen**: Resources; Formal analysis; Supervision; Funding acquisition; Investigation; Visualization; Methodology; Writing—review and editing. **Steffen Thiel**: Conceptualization; Resources; Data curation; Formal analysis; Supervision; Funding acquisition; Validation; Investigation; Methodology; Project administration; Writing—review and editing. **Anne Troldborg**: Conceptualization; Resources; Data curation; Software; Formal analysis; Supervision; Funding acquisition; Validation; Investigation; Methodology; Writing—review and editing. **Markus M Rinschen**: Conceptualization; Resources; Data curation; Formal analysis; Supervision; Funding acquisition; Validation; Investigation; Visualization; Writing—original draft; Project administration; Writing—review and editing.

Source data underlying figure panels in this paper may have individual authorship assigned. Where available, figure panel/source data authorship is listed in the following database record: biostudies:S-SCDT-10_1038-S44318-025-00598-8.

## Funding

## Disclosure and competing interests statement

FD, MMR, ST, and AT are named inventors on a patent application (PCT/EP2025/074596) filed by Aarhus University regarding C-terminal fragments in this paper. MMR received research funding from Novo Nordisk A/S unrelated to this topic. EH acts on the advisory board for Novartis, Morphosys, Sotio, and Argenx.

# Expanded View Figures

**Figure EV1.** **Application of N-termini enrichment to complement inhibition through eculizumab.**

(A) Patient plasma samples with Shiga Toxin producing *E. coli*-induced hemolytic uremic syndrome (STEC-HUS) were treated with anti-C5 antibody (eculizumab), and N-termini quantified after day 8 and day 30 of treatment, compared to baseline conditions. Comparison of N-terminome after day 8 (B), $n = 4$ patients) or day 30 (C), $n = 3$ patients) of eculizumab (anti-C5) treatment vs baseline (day 1). Significantly decreased and increased ($|\log_2 FC| > 1$ & limma moderated t-test *p*-value 0.05) N-termini are depicted in blue and red, correspondingly. Alterations in the N-terminome are more diverse and pronounced on day 8, whereas on day 30, hemopexin cleavages are increasingly present. The C5 N-terminus C5_1381 is significantly up-regulated at day 8 and substantially up-regulated at day 30 of eculizumab treatment (marked in black). (D) Scatterplot of the $\log_2 FC$ N-terminome vs. the $\log_2 FC$ of total proteome after 8 days or 30 days of treatment (E) of treatment. Red data points correspond to complement proteins and show a lower correlation between N-termini and proteome abundance. The corresponding correlation is given as Pearson's correlation coefficient (R) and the significance as paired t-test *p*-values. (F) Significantly regulated complement N-termini ($|\log_2 FC| > 1$ & limma moderated t-test *p*-value < 0.05) during Eculizumab treatment feature a strongly regulated N-terminus for C5 (C5_1381) after day 8 of eculizumab treatment. (G) Structure of complement C5 in complex with the Fab fragment of eculizumab binding to an epitope in the C5/C5b MG7 domain. Gray spheres correspond to the first and last residue of C5. An orange sphere indicates the position of Ile1381. Right, magnified view of the cleavage site proximal to flexible loop in the MG8 domain and the ANA domain released by C5 convertases. Cleavage at position 1381 may lead to partial unfolding of the MG8 domain.

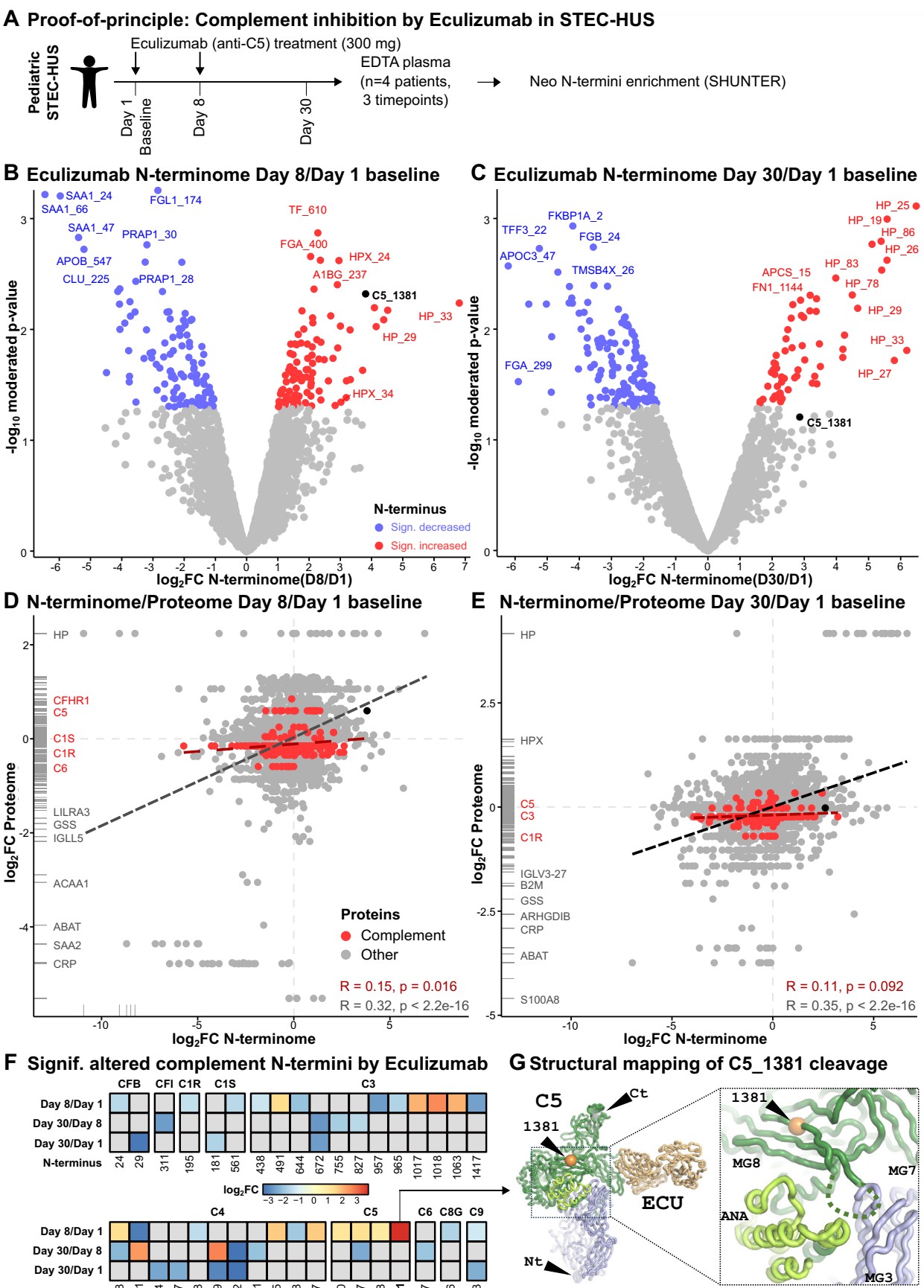

**A** **Proof-of-principle: Complement inhibition by Eculizumab in STEC-HUS**

**B** **Eculizumab N-terminome Day 8/Day 1 baseline**

**C** **Eculizumab N-terminome Day 30/Day 1 baseline**

**D** **N-terminome/Proteome Day 8/Day 1 baseline**

**E** **N-terminome/Proteome Day 30/Day 1 baseline**

**F** **Signif. altered complement N-termini by Eculizumab**

**G** **Structural mapping of C5_1381 cleavage**

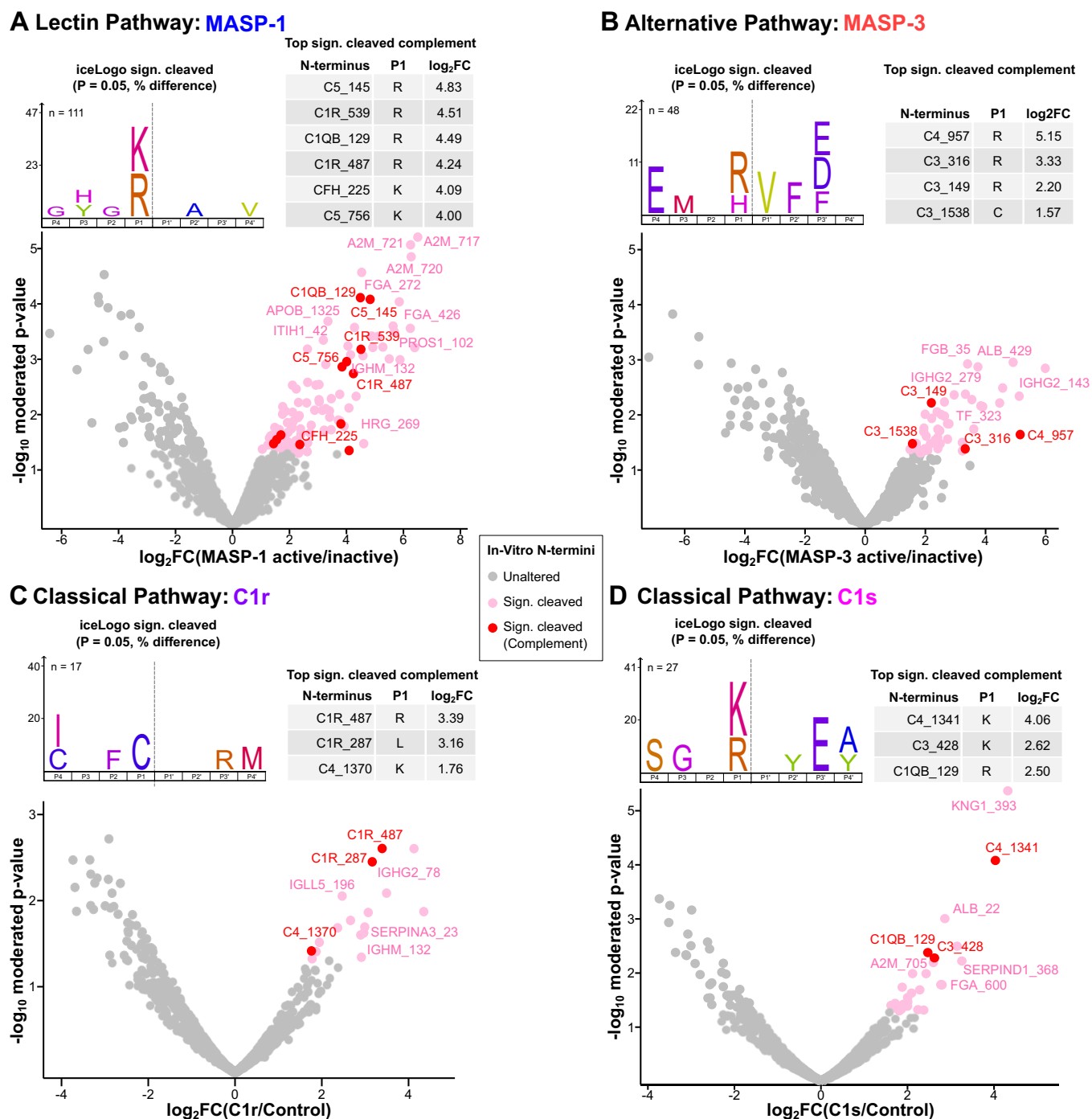

**Figure EV2. In vitro profiling of major human complement initiating proteases, processed with trypsin.**

Volcano plots and motifs of protease-induced proteolytic patterns for the lectin (MASP-1 (**A**)), alternative (MASP-3 (**B**)), and classical pathway (C1r/C1s (**C**, **D**)). After recombinant protease digestion (2 h, 37 °C) and dimethyl labeling, proteins were enzymatically digested using trypsin (Dataset EV10). Substrates were identified by differential abundance (log$_2$FC > 1 & limma moderated t-test *p*-value < 0.05) of active MASP-1/MASP-3 vs. inactive MASP-1/3 or active C1r/C1s vs. no protease control (*n* = 4 individual healthy plasma samples).

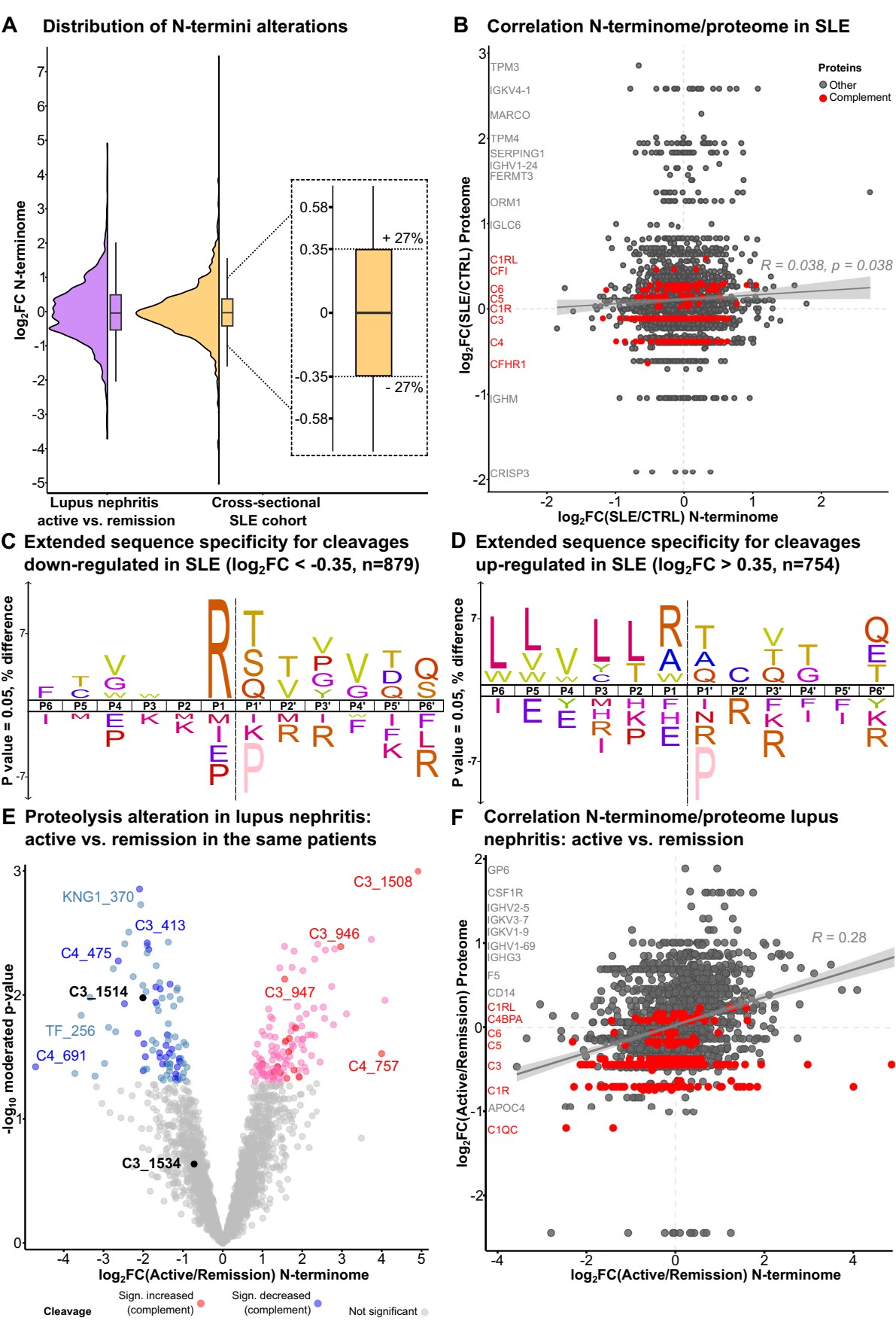

**A** Distribution of N-termini alterations

**B** Correlation N-terminome/proteome in SLE

**C** Extended sequence specificity for cleavages down-regulated in SLE (log₂FC < -0.35, n=879)

**D** Extended sequence specificity for cleavages up-regulated in SLE (log₂FC > 0.35, n=754)

**E** Proteolysis alteration in lupus nephritis: active vs. remission in the same patients

**F** Correlation N-terminome/proteome lupus nephritis: active vs. remission

**Figure EV3. In vivo proteolysis alterations in SLE.**

(A) Distribution of the N-termini alterations for the complete cross-sectional SLE cohort ($n = 166$ samples) and lupus nephritis trajectory ($n = 6$ patients). The first and third quartiles cut-off of $log_2FC$ 0.35 (±27%) is defined as a filter regimen for the subsequent analysis steps of the SLE cohort and is displayed in the zoomed inset (boxplot line represents the median, whereas the box covers the first and third quartiles and the whiskers extend to the 1.5x Inter-Quartile-Range from the box). (B) A high variability of N-termini is detectable in complement proteins (red), which is lacking in the bulk proteome. The overall correlation for the N-terminome and proteome alterations was low (Pearson's R = 0.038). (C) Extended sequence specificity as visualized by iceLogo for the down-regulated N-termini in SLE ($log_2FC < -0.35$, $n = 879$ non-redundant N-terminal cleavage windows; *p*-value = chance of occurrence for every amino acid on every position < 0.05 cut-off used for generation) depicts an arginine-specific motif, mostly followed by glutamine, serine and threonine. (D) Extended sequence specificity of the up-regulated N-termini in SLE ($log_2FC > 0.35$, $n = 754$ non-redundant N-terminal cleavage windows, *p*-value < 0.05 cut-off used for generation) reveals a much more heterogeneous cleavage specificity with arginine in the center, but a remarkable leucine motif ahead of the cleavage site (P6, P5, P3 and P2). (E) In vivo proteolysis alteration in Lupus nephritis patients ($n = 6$ individual patients); the change in the N-termini of in active vs. remission states is displayed. Significantly decreased or increased N-termini ($|log_2FC| > 0.58$ & limma moderated t-test *p*-value < 0.05; Dataset EV9) are illustrated in pale blue and pink, whereas significantly altered complement N-termini are colored in blue and red, respectively. C3-LHF1 and an adjacent cleavage at D1534 are displayed in black. (F) The correlation between the N-terminome and proteome for the lupus nephritis patients was substantially higher (Pearson's R = 0.28). A general increase in the proteome abundance of complement proteins can be observed in the active state, whereas the proteolytic processing appears to differ.

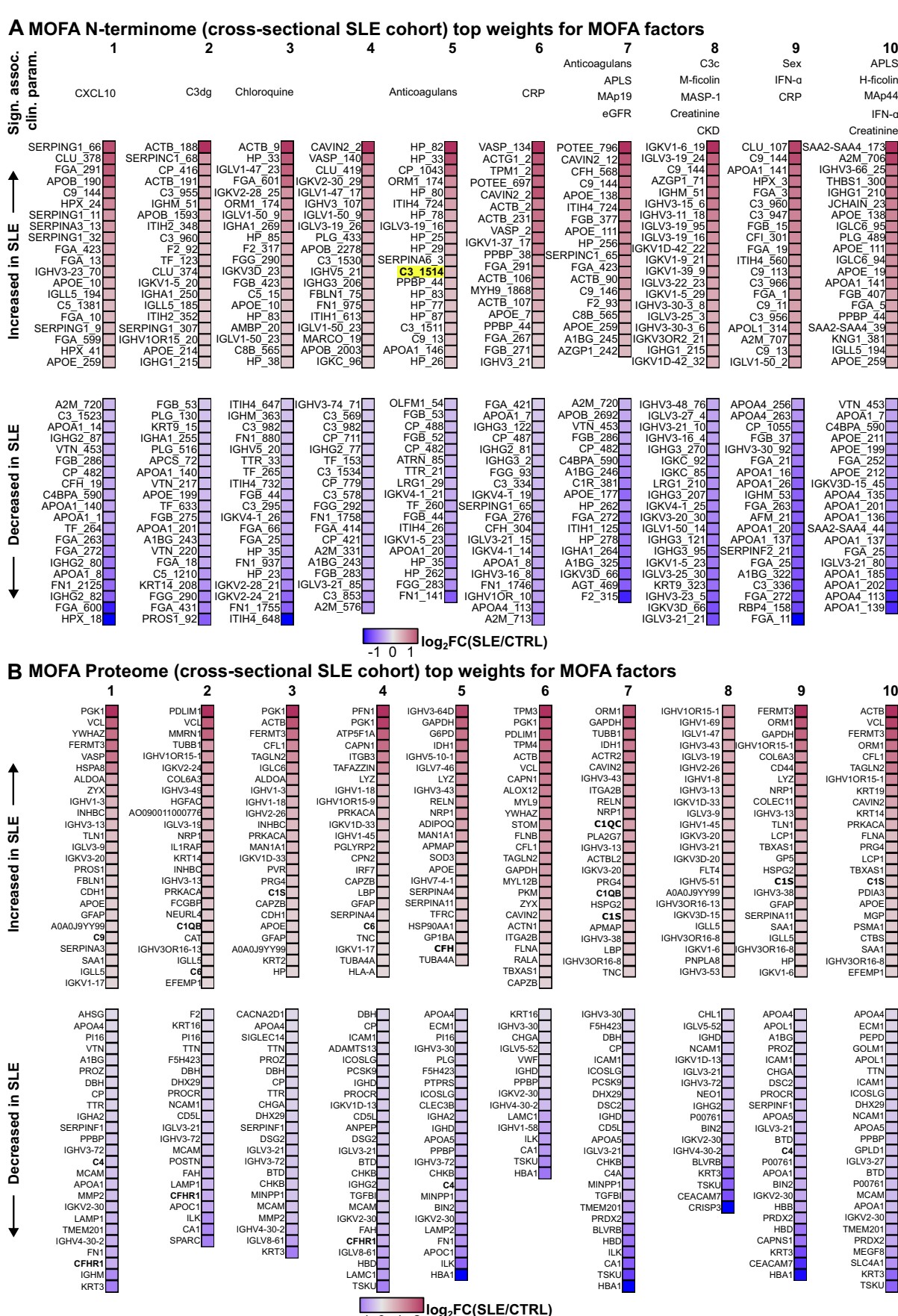

**A** MOFA N-terminome (cross-sectional SLE cohort) top weights for MOFA factors

**B** MOFA Proteome (cross-sectional SLE cohort) top weights for MOFA factors

◀ **Figure EV4. Top N-terminome & proteome weights for all MOFA factors.**

(A) N-terminome-based top positive weights with the most substantial influence on each of the ten MOFA factors were assessed in the cross-sectional SLE patient data, filtered for a minimal $\log_2$FC of 0.35 (27% alteration, cf. Fig. EV3A), and sorted by increased (red) or decreased (blue) abundance in SLE. The candidate N-terminus C3_1534 (C3-LHF1) is prominently present in factor 5 (bold, highlighted in yellow). (B) Proteome-based top positive weights in the cross-sectional SLE patient data are given.

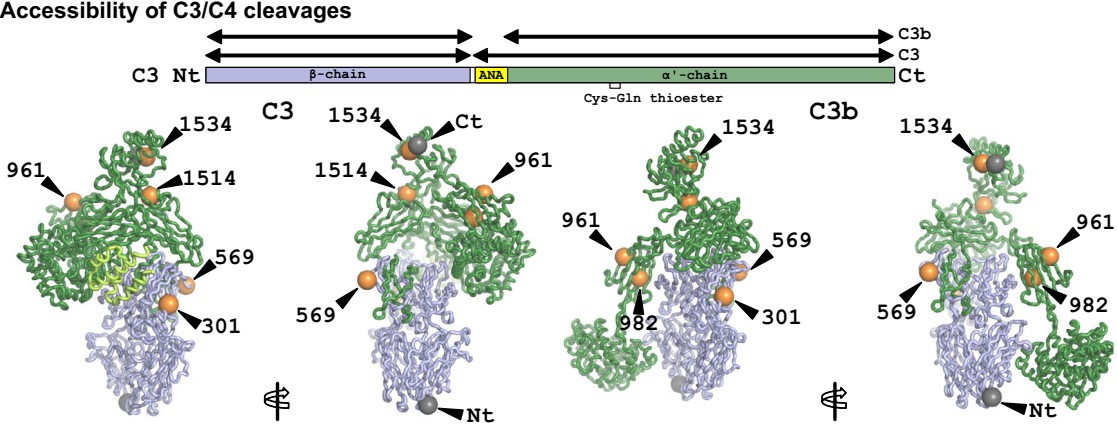

**Figure EV5. Complement C3 N-termini mapping.**

Mapping MOFA-selected C3 cleavages on PDB structures demonstrates accessibility, especially for the C3-LHF1 cleavage (pos. 1514) in C3 and C3b.

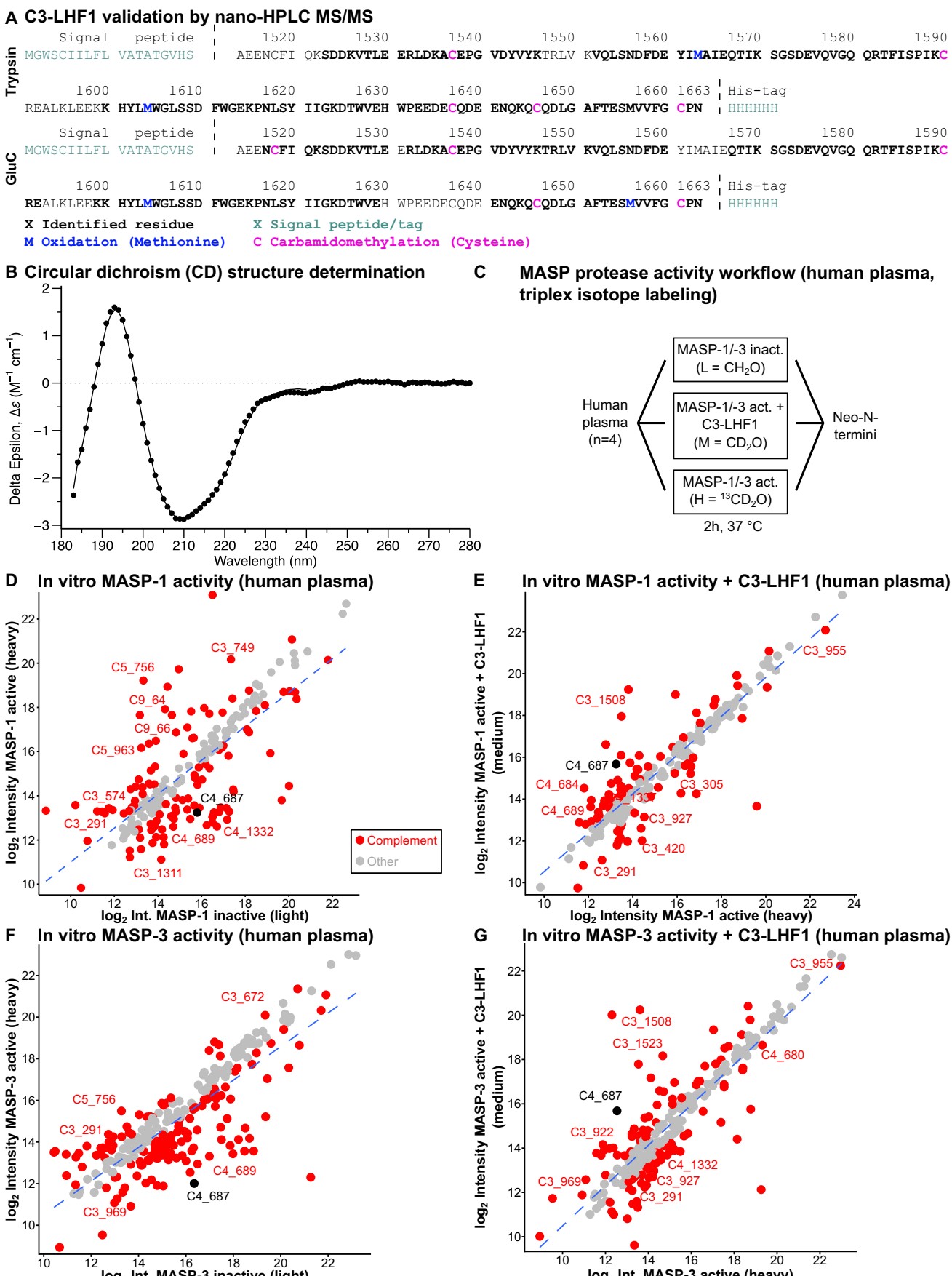

**A C3-LHF1 validation by nano-HPLC MS/MS**

**B Circular dichroism (CD) structure determination**

**C MASP protease activity workflow (human plasma, triplex isotope labeling)**

**D In vitro MASP-1 activity (human plasma)**

**E In vitro MASP-1 activity + C3-LHF1 (human plasma)**

**F In vitro MASP-3 activity (human plasma)**

**G In vitro MASP-3 activity + C3-LHF1 (human plasma)**

**Figure EV6. Sequence and structural characterization of recombinant C3-LHF1.**

(A) Recombinant C3-LHF1 was processed with the two different digestion enzymes, trypsin, and GluC, and subsequently analyzed by nano-LC-MS/MS. Almost full sequence coverage (bold) was achieved in database searches against the host CHO proteome for heterologous protein production, supplemented with the C3-LHF1 sequence. (B) The recombinant LHF-1 protein sample was analyzed using Synchrotron Radiation Circular Dichroism (SRCD) and compared with the crystal structure of the C345c domain within the structure of full-length complement C3 (PDB entry 2A73). The fit of the spectrum resulted in an estimation of 17 ± 3% α-helix, 25 ± 5% β-sheet, 13 ± 4% turns and 39 ± 6% other unordered structures as the secondary structure content. This compares well with the secondary structure content percentages of the crystal structure of the C345c domain of complement C3, obtained via DSSP which is ~28% α-helix, ~33% β-sheet and ~39% other structures. (C) Workflow for determining the MASP-1/3 protease activity in absence/presence of 20 μM C3-LHF1 in a triplex isotope labeling scheme on human EDTA-plasma, where samples after protease incubation were pooled before SHUNTER N-termini enrichment. (D–G) In vitro protease activity was measured in a triplex isotope labeling on human plasma incubated with inactive MASP-1/3 (subsequently labeled with light formaldehyde $CH_2O$), active MASP-1/3 (heavy formaldehyde $^{13}CD_2O$) and active MASP-1/3 + C3-LHF1 (medium formaldehyde $CD_2O$, $n = 4$ individual plasma samples). Complement cleavages (marked in red or black for C4_687) significantly deviating in $\log_2$-transformed N-termini intensity between compared conditions, active protease vs. inactive controls (D, F) or active protease vs. active protease + 20 μM C3-LHF1 (E, G) are labeled with residue number.

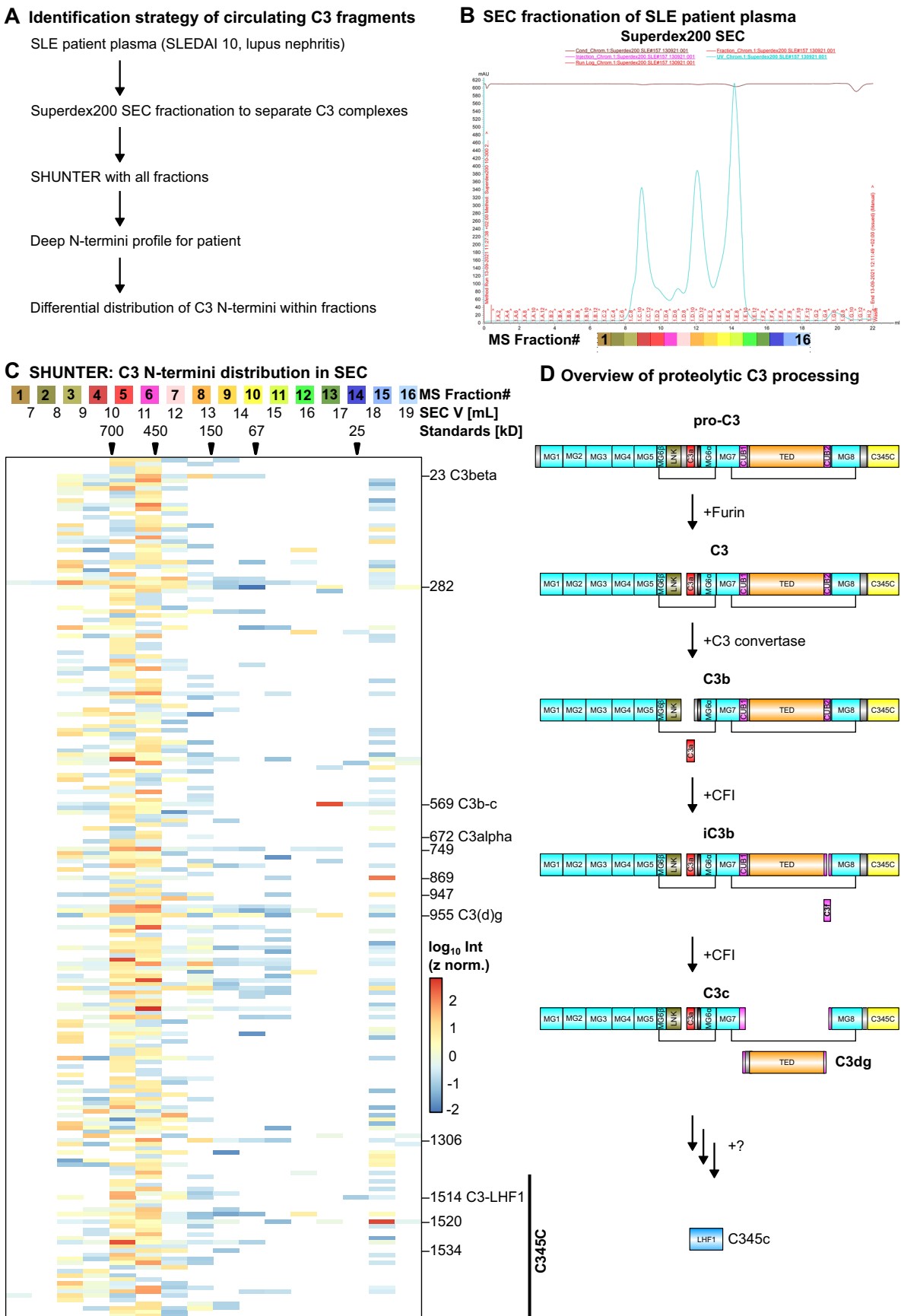

**A** Identification strategy of circulating C3 fragments

SLE patient plasma (SLEDAI 10, lupus nephritis)

↓

Superdex200 SEC fractionation to separate C3 complexes

↓

SHUNTER with all fractions

↓

Deep N-termini profile for patient

↓

Differential distribution of C3 N-termini within fractions

**B** SEC fractionation of SLE patient plasma

**C** SHUNTER: C3 N-termini distribution in SEC

**D** Overview of proteolytic C3 processing

◄   **Figure EV7.   Size exclusion chromatography of SLE patient plasma to identify fragment elution behavior.**

(A) Strategy to identify circulating complement factors by using size-exclusion chromatography (SEC). (B) SEC separation of a severely diseased SLE patient plasma sample (SLEDAI score of 10, LN) yielded sixteen combined 1 mL fractions (1–16), subjected to SHUNTER N-termini enrichment. (C) Heatmap for the overview of the complement C3 N-termini distribution in the SEC MS fractions. The distribution is z-score normalized for each N-terminus, yielding red for a hotspot of the corresponding N-terminus, whereas, in blue labeled fractions, the corresponding N-terminus is only weakly present. C3-LHF1 displays an intermediate distribution profile between C3b (C3_23) and C3(d)g (C3_955) fragments. (D) Schematic overview of the proteolytic processing for complement factor C3.

