## [Peer Review File · The EMBO Journal]

Proteolytic profiling of human plasma reveals an immunoactive complement C3 fragment

Fatih Demir, Elina Kovalenko, Moritz Lassé, Esben Svenningsen, Jens Jensen, Anja Billing, Kathrin Groeneveld, Arvid Hutzfeldt, Lars Nilges, João Guerra, Krzysztof Pietrzak-Lichwa, Yifan Tan, Elizabeth Colby, Annette Hansen, Nazia Kurmasheva, David Olgner, Dongwoo Choi, Mika Richter, Sandra Laufer, Fabian Braun, Sally Johnson, Marcus Krüger, Tobias Huber, Elion Hoxha, Oliver Steinmetz, Ralf Mrowka, Simon Melderis, Moin Saleem, Thomas Poulsen, Gregers Andersen, Steffen Thiel, Anne Trolborg, and Markus Rinschen

Corresponding authors: *Fatih Demir (fatih.demir@biomed.au.dk)* , *Markus Rinschen (rinschen@biomed.au.dk)*

Review Timeline:

Submission Date:	26th May 25
Editorial Decision:	30th Jun 25
Revision Received:	23rd Jul 25
Editorial Decision:	10th Sep 25
Revision Received:	24th Sep 25
Accepted:	30th Sep 25

Editor: Ieva Gailite

Transaction Report:

Dear Dr. Rinschen,

Thank you for submitting your manuscript for consideration by the EMBO Journal. We have now received comments from two reviewers, which are included below for your information. Since the third reviewer was not able to provide their report in a timely manner, I am taking the decision based on the reports at hand.

As you will see, both reviewers are generally positive in their assessment and appreciate the contribution of the study to the research field. While reviewer #1 has no further experimental requests, reviewer #2 with expertise in clinical proteomic and proteolytic analyses indicates a number of technical points that would need further experimental support. Based on these positive assessments, I invite you to submit a revised manuscript in which you address the concerns by reviewer #2. I think that it would be useful to discuss the revision in more detail via email or phone/videoconferencing, in particular regarding the triple radioactive labelling requested by reviewer #2, as the technical feasibility of the proposed analysis is unclear.

We generally allow three months as standard revision time, which can be extended to six months in the case of major revisions. Should you foresee a problem in meeting this deadline, please let us know in advance to discuss an extension.

As a matter of policy, competing manuscripts published during this period will not negatively impact on our assessment of the conceptual advance presented by your study. However, please contact me as soon as possible upon publication of any related work to discuss the appropriate course of action.

When preparing your letter of response to the referees' comments, please bear in mind that this will form part of the Review Process File and will therefore be available online to the community. For more details on our Transparent Editorial Process, please visit our website: <https://www.embopress.org/page/journal/14602075/authorguide#transparentprocess>. Please also see the attached instructions for further guidelines on preparation of the revised manuscript.

Please feel free to contact me if you have any further questions regarding the revision. Thank you for the opportunity to consider your work for publication. I look forward to discussing your revision.

With best regards,

Ieva

Ieva Gailite, PhD
Senior Scientific Editor
The EMBO Journal
Meyerhofstrasse 1
D-69117 Heidelberg
Tel: +4962218891309
i.gailite@embjournal.org

- a point-by-point response to the referees' comments, with a detailed description of the changes made (as a word file).
- a word file of the manuscript text.
- individual production quality figure files (one file per figure)
- a complete author checklist, which you can download from our author guidelines (<https://www.embopress.org/page/journal/14602075/authorguide>).

- Expanded View files (replacing Supplementary Information)

- a Reagents and Tools Table as part of the Methods section, which can be downloaded from our author guidelines

(<https://www.embopress.org/page/journal/14602075/authorguide#structuredmethods>)

We realize that it is difficult to revise to a specific deadline. In the interest of protecting the conceptual advance provided by the work, we recommend a revision within 3 months (28th Sep 2025). Please discuss the revision progress ahead of this time with the editor if you require more time to complete the revisions.

Referee #1:

Assessment:

This reviewer discloses that they had reviewed this current manuscript submitted to EMBO previously for another journal. Thus, my comments are based on this current version and focus on whether the authors had addressed my queries and changes raised in my previous reviews. My previous comments centered largely on method bench-marking (previous HUNTER version), patient classification/clinical parameters, original data presentation (normalized values), general better description of methods and complement and moderating statements about the impact of the c-terminal C3 fragment discovered here on IL-6 biology. This current version satisfies all of my previous requests. Overall, I find the manuscript exciting as it creates not only an important resource but also adds novel insights into the broadening non-canonical activities of C3 and complement in general. I have no further requests for additional experiments.

Referee #2:

The manuscript presents an interesting and significant advancement in N-terminomic profiling through the refined HUNTER technology, providing insights into systemic lupus erythematosus (SLE) and complement biology via serum N-Terminomics. It is also among the first cohort-wide N-terminomic studies. The identification of the novel complement fragment, C3-LHF1, and its immunomodulatory activities is intriguing.

The manuscript would benefit from addressing the following aspects:

Validation of DIA Peptide-Spectrum Matching for Semi-Specific Sequences: While this reviewer personally has confidence in the PSM step, there remains occasional scepticism. Entrapment searches and titration experiments would bolster the reliability of the presented methodology.

Clarification is needed regarding the normalization procedures for the N-terminomic data, especially considering the stated significant per-sample variability. Specifically, it should be addressed whether normalization based purely on post-enrichment abundance might introduce biases, particularly in samples with markedly different N-terminome coverage (e.g., 3% vs. 10% of all digested peptides). Additional experimental data would greatly enhance confidence in these normalization strategies and findings.

Providing a classification of the identified N-termini according to extended amino acid specificity would be useful.

Insight into SLE and complement biology is currently somewhat fragmented and abstract. Restructuring this section to align more with clinical parameters and disease trajectories would improve readability.

The murine stability experiment is promising, yet incomplete incorporation of the C13 isotope potentially complicates interpretation, as the observed ratios reflect mixed signals of synthesis and degradation. Implementing a triple-labeling strategy might improve the interpretability of these data.

The choice of the U2OS cell line for the cell painting assay requires a rationale.

The observed functional and physical interactions with IL6 signaling components represent an intriguing aspect of the study.

We thank the reviewers and editors for their work to make our manuscript better! Please find our responses in bold letters below.

Reviewer 1

Assessment:

This reviewer discloses that they had reviewed this current manuscript submitted to EMBO previously for another journal.

Thus, my comments are based on this current version and focus on whether the authors had addressed my queries and changes raised in my previous reviews. My previous comments centered largely on method bench-marking (previous HUNTER version), patient classification/clinical parameters, original data presentation (normalized values), general better description of methods and complement and moderating statements about the impact of the c-terminal C3 fragment discovered here on IL-6 biology.

This current version satisfies all of my previous requests. Overall, I find the manuscript exciting as it creates not only an important resource but also adds novel insights into the broadening non-canonical activities of C3 and complement in general. I have no further requests for additional experiments.

Response: Thank you for your comments on the previous version, and endorsing the manuscript at this stage.

Reviewer 2

Comment 1: The manuscript presents an interesting and significant advancement in N-terminomic profiling through the refined HUNTER technology, providing insights into systemic lupus erythematosus (SLE) and complement biology via serum N-Terminomics. It is also among the first cohort-wide N-terminomic studies. The identification of the novel complement fragment, C3-LHF1, and its immunomodulatory activities is intriguing.

Response: Thank you for acknowledging both novelty and broad interest of this study!

Comment 2: The manuscript would benefit from addressing the following aspects:

Validation of DIA Peptide-Spectrum Matching for Semi-Specific Sequences: While this reviewer personally has confidence in the PSM step, there remains occasional scepticism. Entrapment searches and titration experiments would bolster the reliability of the presented methodology.

Thank you. While our paper was under review, a high-profile paper was published on controlling FDR in DIA dataset ¹ (published online on June 16 – our paper was submitted on May 26th). This paper suggests that entrapment searches – assessing False detection positives (FDP) using different search strategies such as database shuffling, or foreign species searches, followed by various calculation strategies are necessary to control FDR; particularly protein FDR, and particularly strong in single-cell proteomics datasets – where “true” protein FDR is essentially exploding!

Of course, the continuous development and check of protein and PSM FDR is an important issue. The paper, and also the previous preprint (doi: [10.1038/s41592-025-02719-x](https://doi.org/10.1038/s41592-025-02719-x)) has sparked active discussions in the field, demonstrating increased focus of the continuous development of algorithms, and control of FDR in datasets– see for example <https://github.com/Noble-Lab/FDRBench/issues/6#issuecomment-2987327065> <https://github.com/vdemichev/DiaNN/discussions/1035>. Our laboratory relies on the excellent scholars benchmarking and adjusting search algorithms for FDR control. Currently, however, we cannot see a “gold standard” strategy on how best benchmark FDR in DIA data yet.

Our dataset is chiefly based on peptide PSM. As you write, there are relatively less doubts about the PSM step as compared to protein FDR. Yet, the semi-specific searches in DIA might pose an issue in our Fragpipe/DIA-NN analysis. To this end, we performed the following experiments and analysis to further address the issue as per your suggestion:

1. Entrapment searches

- a. First, we investigated the applicability of *database shuffling approaches*: the mentioned FDRBench tool (<https://github.com/Noble-Lab/FDRBench>) was – at least in our hands – not able to handle isoform databases, and semi-specific free N-terminus searches.
 - i. Our data has consistently been searched against the full human reference proteome from UniProt (canonical + isoforms, 105,707 entries), as is standard to identify true novel proteoforms and N-termini. Unfortunately, the FDR Bench tool could not shuffle this database. Using a canonical-only database (20,453 entries) would be feasible, but it would significantly underestimate our search space.
 - ii. Additionally, the FDR-bench tool (currently) only supports tryptic, ArgC, and completely unspecific shuffling (as relevant for our setup). But all three do not match the characteristics of the purified N-termini: they feature a fixed R at the C-terminus and only a variable N-terminus. Using a completely unspecific digestion approach for our database (> 100 000 entries) was also not feasible due to computational limitations on our side or the FDRBench tool.

For these reasons, we were not able to perform database shuffling as an entrapment search strategy.

- b. Since shuffled search strategies were not applicable, we relied on the combined and lower bound method for FDP estimation as in ¹. Therefore, we performed analyses using an Arabidopsis entrapment strategy to prove that our semi-specific free N-terminus identification strategy within FragPipe is robust. We applied differing peptide and PSM filtering thresholds within FragPipe utilizing three individual human plasma N-termini samples, queried against human canonical + isoforms, reviewed reference proteome database (42 525 entries) supplemented with the reviewed Arabidopsis reference proteome (18 643 entries, $r = 1.438$ for combined database ¹). We communicated with the FragPipe developers and opted for the following settings for the search:

Switching off percolator filtering (min. probability set to 0) and keeping the protein level FDR fixed at 1% (as this setting is not relevant for our peptide-level analysis), we altered the peptide and PSM FDR within FragPipe from 1% to 10%. We could not observe a high entrapment FDP within the scopes or our evaluations, the determined “combined method” and “lower bound” FDP determinations were < 1% until the FDR cut-off was set to 5%. (please see ancillary methods in the appendix for details)

Reviewer Fig. 1: Entrapment searches (human + Arabidopsis thaliana) yielding a low estimated FDP for differing FDR cut-offs within semi-specific free N-terminus searches with FragPipe/DiaNN. With the usually applied 1% cut-off, the FDP is below 1% FDR – only after setting the peptide- and PSM-level FDRs to $\geq 5\%$, the level of bait identifications within Arabidopsis increases over the 1% FDP threshold.

2. Titration experiments.

- a. We also performed titration experiments, spiking in E-coli protein into human plasma at different ratios (1:4, 1:2, 1:1, 2:1 and 4:1) and performing N-terminome analyses. We found that, without normalization, there was a clear increase in the total human N-termini intensity, but the nature of the identifications or the quantification of the human N-termini was not altered, suggesting that the identification is valid (Reviewer Fig. 2).

Reviewer Fig. 2: Robustness of the human N-termini quantifications with DiaNN/FragPipe were assayed with the help of human plasma samples ($n=3$ individual healthy controls) with alternating E.coli N-termini spike-in ratios (4:1 to 1:4 E.coli to human plasma). **a**, Assayed in a non-normalized manner with DiaNN, increasing E.coli spike-ins lead to reduced ratios comparing against the control. non-spiked control measurements. **b**, Comparing the DiaNN quantifications (normalized) for the human N-termini in the 4:1 E.coli:human plasma spike-in with the non-spiked human plasma reveals a robust quantification of the human N-termini regardless of a 4:1 spike-in of E.coli (Pearson's correlation coefficient of 0.94). **c**, Pearson's correlation coefficients for the 4:1 to 1:4 E.coli:human plasma N-termini spike-ins reveal a very high degree of correlation for the spike-in and no spike-in control sample quantifications for the human plasma N-termini.

In total, we believe that it is fair to say that semispecific (N-terminomic) searches have similar behavior as specific searches on peptide FDR level, and identification and quantification is robust even in the presence of a excess of E coli proteome.

Comment 3: Clarification is needed regarding the normalization procedures for the N-terminomic data, especially considering the stated significant per-sample variability. Specifically, it should be addressed whether normalization based purely on post-enrichment abundance might introduce biases, particularly in samples with markedly different N-terminome coverage (e.g., 3% vs. 10% of all digested peptides). Additional experimental data would greatly enhance confidence in these normalization strategies and findings.

We apologize that we did not perform a detailed description of our normalization strategies for this datatype, and we added this now.

First, we used the same protein amount for all HUNTERS (250ug per sample). That suggests that the total starting material is identical for all samples. Then, we performed the sHUNTER protocol and purified the N-termini. We observed that the protein content in the N-termini-enriched samples post-enrichment did not differ among the control and SLE samples.

In this case, it could remain an issue whether a normalization was performed based on total ion intensity pre-enrichment or post-enrichment. We realize that this was not stated in the text. In our setting, we used the default settings of DiaNN which was used for all analyses. This setting performs a normalization step based on ion total intensities post-enrichment. (There are many protein-based normalization steps here, but they do not affect our data.)

To create a use case to see if the normalization may introduce biases, we compared the total DiaNN intensities per N-termini from a human plasma samples with a variable amount of E.coli proteome spike in – with and without normalization. Despite the differing proteome ratios, we see that normalization is very effective, and does not induce significant biases to a sample that has no spike-in.

Reviewer Fig. 3: Robustness of human plasma N-termini quantifications ($n=3$ individual samples) with or without normalization during DiaNN quantification. *E. coli* spike-ins were performed at different ratios of 4:1 to 1:4 on peptide level. **a**, Distribution of human plasma N-termini quantification ratios against no *E. coli* spike-in control (non-normalized). **b**, Excess *E. coli* does not affect quantification of the human plasma N-termini (ratio 1:4 human plasma:*E. coli*, non-normalized, Pearson's correlation coefficient $R = 0.94$). Decreasing amounts of *E. coli* does not alter the correlation of the N-termini quantifications (**c-e**), also at low amounts of *E. coli* (Ⓜ ratio 4:1, non-normalized, $R = 0.959$). **g**, Distribution of the measurements, queried with DiaNN normalization displays the capabilities of DiaNN to normalize the human plasma N-termini quantifications even in higher *E. coli* spike-in ratios like 1:4 (**h**) or 1:2 (**i**). Quantifications are stable throughout the tested 1:1 (**j**), 1:2 (**k**), 1:4 (**h**) and 4:1 (Ⓜ) ratios with a high degree of correlation (**h**, $R = 0.937$ for 1:4 and **l**, $R = 0.961$ for 4:1).

Biologically, we also do not see that normalization has an effect, at least on the effect of SLE in our patients (Reviewer Fig. 4: Regulation in the SLE and CTRL samples shows a high degree of correlation for normalized/non-normalized ratios, thus the effect of the DiaNN-inherent normalization has no substantial effect on the quantification. Reviewer Fig. 4).

Reviewer Fig. 4: Regulation in the SLE and CTRL samples shows a high degree of correlation for normalized/non-normalized ratios, thus the effect of the DiaNN-inherent normalization has no substantial effect on the quantification.

Regarding the reviewer's specific comment regarding N-termini coverage, we have calculated the mean protein N-termini sequence coverage as the peptide sequence coverage. We can observe a quite homogeneous distribution among the SLE cohort with

a peak N-termini coverage of ~11% for all proteins per sample.

Reviewer Fig. 5: Overall mean N-termini coverage distribution for SLE cohort samples.

Investigating the individual protein N-termini coverage and the corresponding regulation of the N-terminus in SLE (Reviewer Fig. 6a) or the general abundance (as \log_{10} intensity in Reviewer Fig. 6b) did not show any pronounced effect of the N-termini coverage on the interpretation for our N-termini data. Especially the regulation remains not affected by the individual protein N-termini coverage, whether DiaNN normalization was applied or not (Reviewer Fig. 6a).

Reviewer Fig. 6: The protein N-termini coverage in the SLE cohort was mapped to the regulation and intensities. **a**, No correlation can be observed for the protein N-termini coverage with the regulation in $\log_2FC(SLE/CTRL)$ – regardless of if normalized or non-normalized. **b**, There is no correlation for the \log_{10} intensities with the protein N-termini coverage, which is comparable among the CTRL and SLE samples and does also not differ greatly among the normalized or non-normalized samples.

Comparing the CTRL and SLE samples in general, we could not observe general differences in the degree of N-termini coverage (Reviewer Fig. 7).

Reviewer Fig. 7: The mean protein N-termini coverage did not differ among the SLE and CTRL samples (Pearson's correlation coefficient of 1.0).

Additionally, we inspected whether the distribution has any effect on the regulation in SLE by classifying the N-termini coverage into quantiles and comparing the quantifications (Reviewer Fig. 8). The $\log_2FC(SLE/CTRL)$ was not statistically different.

Reviewer Fig. 8: The protein N-termini coverage was classified into quantiles (Q1: 0-18%, Q2: 19-30%, Q3: 31-36%, Q4: 37-56%, Q5: 67-81%) and DiaNN N-termini quantifications were determined – no significant difference (p-values for a two-sided student's t-test are given) was observed between the quantiles, regardless if DiaNN N-termini quantifications were normalized or not.

In conclusion, while the reviewer is certainly correct that pre- or post-enrichment normalization strategies are important, we here believe that we have mitigated this issue by (1) using same amount of input material on proteome level (250 µg for all samples), and (2) showing that normalization does not have an overall effect on the biological effect of SLE in our patients, and (3) demonstrating that the N-termini coverage does not introduce a significant bias to the quantification.

Comment 4: Providing a classification of the identified N-termini according to extended amino acid specificity would be useful.

We ran a motif analysis to classify the identified N-termini according to extended amino acid specificity (Reviewer Fig. 9). These iceLogos have been added to Suppl./EV Fig. 4 as panels c & d.

Reviewer Fig. 9: Extended cleavage specificities as rendered iceLogos, derived from non-redundant N-terminal cleavage windows from the up- and down-regulated N-termini in SLE ($\log_2FC > 0.35$ or < -0.35 ; number of respective cleavage windows is indicated in parentheses).

Comment 5: Insight into SLE and complement biology is currently somewhat fragmented and abstract. Restructuring this section to align more with clinical parameters and disease trajectories would improve readability.

We acknowledge the reviewer's point regarding clarity. We have revised the SLE section of the introduction to more directly align with clinical disease features, including disease activity and lupus nephritis, to better contextualize the proteolytic signatures.

Comment 6: The murine stability experiment is promising, yet incomplete incorporation of the C13 isotope potentially complicates interpretation, as the observed ratios reflect mixed signals of synthesis and degradation. Implementing a triple-labeling strategy might improve the interpretability of these data.

Thank you for bringing this up. We and others²⁻⁴ have used double labeling as approach to approximate the "age" of protein or an N-terminus – disregarding if they are synthesized or degraded.

However, we completely agree that formally, a triple labeling strategy will maximize the chances to disentangle synthesis and degradation. This has been elegantly demonstrated in cell culture systems, for instance in⁵. However, here we have an in vivo situation – this means that interorgan handling and recycling of amino acids contribute to a large degree as well (for instance, lysine is also synthesized by the gut microbiome!⁶). This means, that kinetic calculations are not very straightforward in vivo as compared to in vitro.

Labeling a cohort of mice, particularly in the triple-labeling setting would be a years-long project, requiring some 10s of thousands of Euro in costs for the stable isotopes. We – in coordination with the editor – considered this not feasible in the timeline of a revision. We have acknowledged the limitation mentioned by you clearly in the text now.

Comment 7: The choice of the U2OS cell line for the cell painting assay requires a rationale.

We selected the human osteosarcoma line U-2 OS because it provides a biologically and technically clean platform with reference data to maximize the chances of discovering bioactivity.

First, U2OS have no large T antigen or viral infection compared to many other cell line (consequently basal antiviral signaling is low). Further, they respond to a variety of

immune stimuli. Relevance in complement biology has been reported to these cells ⁷, and links between complement system and cancer exist ⁸⁻¹⁰.

There are also technical reasons that favor U2OS cell lines over other cell lines in cell painting. First, the cell line grows easily and reproducibly, and are easy to stain. Second, there are extensive reference data available. The JUMP Cell painting dataset (Broad Institute) contains cell painting data from more than 116750 compound perturbation ¹¹ – all for U2OS cells. Also we had more than reference profiles from hundreds compounds available in house ¹². Interestingly, the cell painting profile was not straightforwardly linked to any known compounds, suggesting that it is really a separate signaling mechanism.

Comment 8: The observed functional and physical interactions with IL6 signaling components represent an intriguing aspect of the study.

Thank you!

Ancillary Methods

Entrapment searches

Three individual healthy human control EDTA-plasma samples were used to perform a SHUNTER N-termini enrichment with each 250 µg of starting proteome. Labeling, methodology and mass spectrometry were done exactly as for all the other SLE samples. Data analysis was performed with FragPipe v23.0 and the build-in DiaNN 1.8.1-beta2 and the following modifications to the normal data analysis workflow in the manuscript:

1. Percolator validation was switched off by setting the threshold to 0
2. For the validation within FragPipe, protein-level FDR was fixed to 0.01 (1%) as variations in the protein FDR had no influence on peptide-level identifications (“--prot 0.01”)
3. Peptide and PSM-level FDR were varied from 0.01 (1%) to 0.1 (10%) by using the corresponding arguments (“--pep 0.01 --psm 0.01”)
4. Total number of identified peptides and entrapped *Arabidopsis thaliana* peptides were counted from the peptides.tsv output file within FragPipe

E.coli spike-in experiments

In parallel to the three individual human EDTA-plasma samples, an E.coli cell pellet (0.48g Fw) was lysed in 2% NP-40, 50 mM HEPES pH 7.4, 0.1M NaCl, 2.5 mM EDTA, Roche cComplete protease inhibitors for 1h at 4 °C and centrifuged at 13 000 g, 4 °C for 30 min. to pellet debris. The protein content in the lysate was determined via BCA and a big E.coli N-termini sample was generated in parallel to the human plasma samples. After N-termini enrichment, the peptide content of the purified N-termini was determined by NanoDrop A₂₈₀ and corresponding pools with in total 20 µg of N-termini and different spike-in E.coli N-termini amounts were generated in 1:4 to 4:1 ratios. From each mixture, 1 µg was injected onto the Exploris480 nLC/MS mass spectrometer setup and measured and analyzed as for the SLE cohort samples. Data analysis was carried out with FragPipe v23.0 and the build-in DiaNN 1.8.1-beta2 with default settings for ¹³CD₂O-labelled N-termini (semi-specific free N-terminus, ArgC specificity, fixed dimethyl on K and peptide N-terminus and fixed carbamidomethyl on C; variable oxidation on M) and with switched off normalization (“--no-norm” option enabled).

References

1. Wen, B. *et al.* Assessment of false discovery rate control in tandem mass spectrometry analysis using entrapment. *Nat Methods* **22**, 1454–1463 (2025).
2. Rinschen, M. M. *et al.* A Multi-layered Quantitative In Vivo Expression Atlas of the Podocyte Unravels Kidney Disease Candidate Genes. *Cell Rep* **23**, 2495–2508 (2018).
3. Li, W. *et al.* Turnover atlas of proteome and phosphoproteome across mouse tissues and brain regions. *Cell* **188**, 2267–2287.e21 (2025).
4. Lang, F. *et al.* Dynamic changes in the mouse skeletal muscle proteome during denervation-induced atrophy. *Dis Model Mech* **10**, 881–896 (2017).
5. Boisvert, F.-M. *et al.* A quantitative spatial proteomics analysis of proteome turnover in human cells. *Mol Cell Proteomics* **11**, M111.011429 (2012).
6. Tan, Y., Chrysopoulou, M. & Rinschen, M. M. Integrative physiology of lysine metabolites. *Physiol Genomics* **55**, 579–586 (2023).
7. Jeon, H. *et al.* Activation of the complement system in an osteosarcoma cell line promotes angiogenesis through enhanced production of growth factors. *Sci Rep* **8**, 5415 (2018).
8. Yoneda, M. *et al.* Enhancement of cancer invasion and growth via the C5a-C5a receptor system: Implications for cancer promotion by autoimmune diseases and association with cervical cancer invasion. *Oncol Lett* **17**, 913–920 (2019).
9. Merle, N. S. & Roumenina, L. T. The complement system as a target in cancer immunotherapy. *Eur J Immunol* **54**, e2350820 (2024).

10. Saxena, R. *et al.* Complement factor H: a novel innate immune checkpoint in cancer immunotherapy. *Front Cell Dev Biol* **12**, 1302490 (2024).
11. Chandrasekaran, S. N. *et al.* JUMP Cell Painting dataset: morphological impact of 136,000 chemical and genetic perturbations. 2023.03.23.534023 Preprint at <https://doi.org/10.1101/2023.03.23.534023> (2023).
12. Svenningsen, E. B. & Poulsen, T. B. Establishing cell painting in a smaller chemical biology lab – A report from the frontier. *Bioorganic & Medicinal Chemistry* **27**, 2609–2615 (2019).

Dear Markus,

Thank you for submitting a revised version of your manuscript. We have now received input from one of the original reviewers, who is satisfied with the provided revisions. According to the reviewer's request, please add the indicated data in the Appendix for an easier access for the broader readership. Additionally, there are a few editorial and formatting aspects that need to be addressed before I can extend official acceptance of the manuscript:

1. Please submit up to five keywords for your manuscript.
2. Please check if the email provided for the co-author Sally A. Johnson (sally.johnson@nuth.nhs.uk) is correct.
3. Please check that the funding information is correct and identical both in the manuscript and our online system; currently, multiple grants are not listed in our submission system. In the manuscript text, please add funding information to the Acknowledgments section.
4. Please add legends for figures and Expanded View (EV) figures at the end of the manuscript after References.
5. Please remove figures from the manuscript text file and provide only as individual, production quality files.
 1. There are currently 12 EV figures, while we usually recommend up to 5-7 EV figures. Please consider moving some of the EV figures to the Appendix. The Appendix Figures would need to be renamed Appendix Figure S1 etc. and would need to be compiled together with their legends into a single PDF file labelled "Appendix", which would also need a brief table of contents. The pages should be numbered.
6. CRediT has replaced the traditional author contributions section because it offers a systematic, machine-readable author contributions format that allows for more effective research assessment. Please remove the Authors Contributions from the manuscript and use the free text boxes beneath each contributing author's name in our online submission system to add specific details on the author's contribution. More information is available in our guide to authors.
7. Please rename "Conflict of interest" section into "Disclosure and competing interests statement" (further info: <https://www.embopress.org/page/journal/14602075/authorguide#conflictsofinterest>).
8. Please rename "Data sharing statement" section into "Data Availability" and move to the end of "Methods" section. Please add a resolvable link to the proteomics dataset.
9. Please update references according to The EMBO Journal style - where there are more than 10 authors on a paper, the first 10 should be listed, followed by 'et al.' DOIs should only be used for preprints and datasets that have not been published yet. Please see further information here: <https://www.embopress.org/page/journal/14602075/authorguide#referencesformat>
10. Please rename Supplementary Tables 1-21 into Dataset EV1-21 and update the callouts and legends accordingly. Please remove the legends from the manuscript text file and add to each dataset file in a separate tab/sheet.
11. All Materials and Methods need to be described in the main text using our 'Structured Methods' format. According to this format, the Methods section includes a Reagents and Tools Table (listing key reagents, experimental models, software and relevant equipment and including their sources and relevant identifiers) followed by a Methods and Protocols section describing the methods, ideally using a step-by-step protocol format. The aim is to facilitate adoption of the methodologies across labs. Please download and fill our Reagents and Tools Table template (.docx), which you can find in our author guidelines: <https://www.embopress.org/page/journal/14602075/authorguide#structuredmethods>
When submitting your revised manuscript, please do not include the Reagents and Tools Table in the Methods section of the manuscript but upload it as a separate file choosing the file type "Reagent Table".
An example of a Method paper with Structured Methods can be found here: <https://www.embopress.org/doi/10.15252/msb.20178071>.
12. Our data editors have flagged the following issues in figure legends that need correcting:
 - Please define the annotated p values ****/***/**/* as well as provide the exact p-values for the same in the legend of figure 6D, F; EV11 D as appropriate.
 - Please provide the exact p values in the legends of figures 4D, 7E; EV2 B, C, D, E; EV3 A-D; EV4 E; EV11 C, EV12 A, B.
 - Please indicate the statistical test used for data analysis in the legends of figures 2F, 6D, F; 7E, F; EV2 B, C, D, E; EV3 A-D; EV4B, C, D, E; EV8 A, B; EV10B, C; EV11 C, D; EV12 A, B.
 - Please note that the box plots need to be defined in terms of minima, maxima, centre, bounds of box and whiskers, and percentile in the legend of figure 5B.
 - Please define the box plots in terms of minima, maxima, centre, and percentile in the legend of figure EV4 A.
 - Please provide information on the number and nature of replicates in the legends of figures 5B, EV1 C; EV2 B, C; EV4 E, EV10B, EV11 C, D.
 - Please define the error bars in the legends of figures EV1 B, C; EV11 C, D.
 - Please define the measure of center for the error bars in the legends of figures 4D, E; 6D-F; 7E; EV12 A, B.
 - Please define the scale bar needs to be defined for figure EV11 B.
13. Papers published in The EMBO Journal are accompanied online by a 'Synopsis' to enhance discoverability of the manuscript. It consists of A) a short (1-2 sentences) summary of the findings and their significance, B) 3-4 bullet points highlighting key results (the highlights can be repurposed for this) and C) a synopsis image that is 550x300-600 pixels large (width x height, jpeg or png format). You can either show a model or key data in the synopsis image. Please note that the image size is rather small and that text needs to be readable at the final size.

With kind regards,

Ieva

We realize that it is difficult to revise to a specific deadline. In the interest of protecting the conceptual advance provided by the work, we recommend a revision within 3 months (9th Dec 2025). Please discuss the revision progress ahead of this time with the editor if you require more time to complete the revisions.

Referee #2:

The authors have done a fantastic job addressing this reviewer's suggestions.

Many of the "reviewer figures" in the rebuttal letter provide very important insight into the robustness and applicability of N-terminomics. I have not spotted these figures in the supplementary material and hope not to have overlooked. I urge the authors to include their "reviewer-only" data on N-terminomics, identification, and normalization as supplementary material - this will be very valuable data for the entire proteomic and N-terminomic community.

I look forward to seeing this work being published and congratulate the authors on a very nice work!

The authors addressed the remaining editorial issues.

Dear Markus and Fatih,

Thank you for addressing the final editorial points. I apologise for the slow process from our side due to the high number of submissions that we experience at the moment. I am now pleased to inform you that your manuscript has been accepted for publication - congratulations!

Before we forward your manuscript to our publishers, I would like to propose some minor edits in the manuscript abstract and synopsis (please see below and in the attached file). I have also written a short blurb that will accompany the title of your manuscript in our online table of contents. Please take a look and let me know if any corrections or adjustments are needed.

Blurb:

A resource of protease-generated protein fragments in human plasma links complement signatures to clinical outcomes in autoimmune disease and uncovers a C-terminal fragment of C3 that activates IL6ST cytokine receptor.

Synopsis:

Activation of the complement cascade of proteases has been linked to autoimmune and kidney diseases, including systemic lupus erythematosus (SLE). Here, optimization of a workflow for protease activity profiling reveals a highly abundant, stable C-terminal fragment of complement component 3 (C3) with an immunomodulatory effect.

- Development of a scalable workflow for plasma N-terminomics allows profiling of the proteolytic properties of major complement proteases and of half-life-dependent protein processing.
 - Mapping of SLE-patient proteolytic fingerprints links complement fragment signatures to clinical outcomes including renal function, inflammation, and therapy response.
 - Non-canonical MASP proteolysis generates a highly abundant, stable C-terminal fragment of C3, C3-LHF1, which inhibits classical and MBL complement pathways.
- C3-LHF1 binds to and activates IL6ST (gp130), altering immunomodulatory JAK/STAT3 signaling in human kidney organoids.

If you have any questions, please do not hesitate to contact the Editorial Office. Thank you for this contribution to The EMBO Journal and congratulations on an exciting study!

With best wishes,

Ieva
